


# Sensitivity analysis of input ground motion on surface motion
# parameters in high seismic regions: A case of Bhutan Himalaya
Karma Tempa[1], Komal Raj Aryal[2], Nimesh Chettri[1], Giovanni Forte[3], Dipendra Gautam[4, *]
[1]Civil Engineering Department, College of Science and Technology, Royal University of Bhutan,
Phuentsholing, Bhutan
[2]Faculty of Resilience, Rabdan Academy, Abu Dhabi, United Arab Emirates
[3]Department of Civil, Environmental and Architectural Engineering (DICEA), University of Naples Federico II,
Naples, Italy
[4]Department of Civil Engineering, Institute of Engineering, Thapathali Campus, Kathmandu, Nepal
[*] *Correspondance*: Dipendra Gautam (dipendra01@tcioe.edu.np)
**Abstract.** Historical earthquakes demonstrated that the combination of the characteristics of strong ground motion
and local soil conditions significantly influence the seismic site response of a particular site. Since there are no
instrumental records of past earthquakes in Bhutan, this study is the first attempt to quantify the influence of the
local site conditions in the eastern fringe of the Himalaya considering various global historical earthquakes on a
deterministic basis. According to the recent Global Seismic Hazard Map (GSHAP), Phuentsholing Thromde (city)
in Bhutan is likely to be exposed to the peak ground acceleration (PGA) between 0.20 g – 0.28 g. This particular
scenario highlights the likelihood of occurrence of stronger shaking; considering the proximity of active seismic
faults. To this end, we performed one-dimensional (1D) ground response analysis in eight different locations
considering earthquakes with a wide variation of PGA even beyond 0.2-0.28g so as to quantify the possible
seismic response of soil strata under wide range of ground shaking. Sensitivity analysis is performed by a
statistical correlation function to correlate the ground motion parameters for different earthquake shaking
intensities. The amplification responses of the local soil from each input motion are projected to predict seismic
hazard scenarios that highlight the ground shaking response for some locations in Phuentsholing, Bhutan, which
is one of the densest urban centers in the country. The study highlights the critical range of the fundamental natural
period roughly between 0.9 s to ~ 5.0 s with the highest range of seismic wave amplification between ~ 2.8 to 6.2
due to the local ground conditions, suggesting a likely aggravation during earthquakes that may lead to severe
consequences especially in terms of infrastructure damages.
**Keywords:** seismic site effect, amplification factor, soil fundamental period, sensitivity analysis, Bhutan.
## 1   Introduction
Bhutan Himalaya is an area vulnerable to multiple kind of natural hazards among them earthquake is one of the
most devastating  for the country. Historical records highlight anomalous damage pattern in various parts of the
country. Thus, there is a strong likelihood of seismic site effects on structure and infrastructure damage especially
in the plain areas of the country. In Bhutan, very few studies on local seismic response analysis have been
conducted so far. Some of the recent studies reveal impacts of bedrock depth to ground response parameters
including projection of engineering bedrock through shear wave profiling to study the similar effects on ground



responses e.g., (Tempa et al., 2020; Tempa, Chettri, Gurung, et al., 2021). However, these studies considered a
single strong ground motion in which ground response analysis may be detrimental to adequately and accurately
delineate the response parameters (Stevens et al., 2020). Provided that the seismicity of the study area is high due
to fragile seismotectonic and geological settings, the sites may experience earthquakes of different magnitudes in
the future. To comprehend such possibilities, we aim to bridge the gap by using six typical earthquakes to
demonstrate the variability of ground response through correlation and sensitivity analysis.

42        Himalaya is one of the most seismically active regions in the world. Historical earthquakes that occurred in

this region have resulted in enormous losses and damages (Gautam et al., 2016) and thus the impending
earthquakes are certain to strike the region with detrimental consequences. The eastern fringe of Himalaya, i.e.,
Bhutan, and neighboring areas were strongly affected by significant earthquakes, records of past earthquakes in
Bhutan are available since 1713 (Mw 7.0) however, most of those occurred in the 18[th] century are not well-
documented. Recently, on April 05, 2021 a $M_w$ 5.0 earthquake  occurred in southern Bhutan in Samtse and on
September 2009 Mongar earthquake (Mw 6.7) caused major damages in the eastern parts of Bhutan and
considerably affected other parts of the Country (Chettri et al., 2021b). Local site conditions during historical
earthquakes in Bhutan were identified as the main cause of structural damage. For this reason, this study attempts
to quantify the characteristics and effects of different strong ground motions to seismic responses in the area under
consideration. Seismic ground response analysis plays a crucial role, and the method is widely used by researchers
for various applications that can capture local ground effects or site conditions, especially when estimating and
predicting ground motion variation (Chavez-Garcia et al., 1990). For several decades, the approach has been
followed to address seismic hazard assessment and mapping, which includes microzoning liquefaction
susceptibility assessment in addition to the studies conducted on local sites effects by Lopez-Caballero et al.,
(2012), Gautam and Chamlagain (2016), and Sil and Haloi (2018), among others. The outcomes of such studies
aim to provide local seismic hazard parameters which are imperatively utilized by structural engineers for
earthquake-resistant design of infrastructures at the particular site or location (Douglas, 2006). The ground motion
characteristics influence seismic response attributes largely. The expected level of seismic ground motion as an
input parameter provides a quantitative approximation that is not only employed in earthquake-resistant design
but also, other accompanying earthquake hazards such as landslide, liquefaction, and seismic risk assessment
(Bommer and Martinez-Pereira, 2000). These ground response parameters typically characterize the complex
nature of strong-motion accelerograms using a simple expansion of predictive relationships. The two prominent
deterministic and probabilistic approaches are widely used for seismic hazard studies globally. Wyss and Rosset
(Wyss and Rosset, 2013) stated that the standard probabilistic seismic hazard assessment method (PSHA) leads
to an over or underestimation of the expected acceleration and intensity in areas with low and high seismicity
often resulting in incorrect results. Similarly, Tempa, Chettri, Gurung, et al. (2021) recommend the use of a
deterministic approach that can calculate the accelerations and losses that would occur if the maximum considered
earthquake (MCE) occurs. In addition, selecting a single ground motion by considering only amplitude for seismic
hazard analysis may not be a reliable approach to estimating site amplification. Hence, the ground motion
parameters that are related to the amplitude are investigated to examine and predict the variability, often
considered sensitivity concerning mean values and associated scatter. Although input motion selection is a
complex procedure, a simple approach widely adopted is to is to scale ground motion records to  a target spectral
acceleration in the fundamental period of the structure of interest (Kramer et al., 2012). To comprehend such



basis, the initial preliminary study aimed at covering investigation on building typologies in Phuentsholing city,
which is mainly composed of buildings up to eight storey. The basis of the selection of input motions was also
established in connection with the design spectra which confirm the Indian standard code IS 1893: 2002 (IS:1893,
2002). In particular, based on the detailed geotechnical test data and shear wave velocity profiles developed by
(Tempa, Chettri, Gurung, et al., 2021), the sites are further investigated for the variability of seismic response to
six different input ground motions. In this paper, sensitivity analysis of site response for specific soil conditions
in Phuentsholing, (Bhutan) is explored by a statistical correlation function to correlate the ground motion
parameters for different earthquake shaking intensities. The study area is very significant, as Phuentsholing is one
of the major urban and commercial hubs in Bhutan Himalaya and seismic site effects on existing structures may
have detrimental consequences due to inherent vulnerabilities of structures and infrastructures as well as presence
of loose soil deposits. To quantify the seismic site effects, a range of time histories is selected and site response
parameters are estimated.

## 2 Seismicity and geological setting of the study area

The Himalayan region is one of the most seismically active zones characterized by both large and moderate-sized
events (Drukpa et al., 2006). Bhutan is located in the eastern Himalayas formed due to the subduction of the Indian
plate beneath the Eurasian Plate and spans from the low-lying Brahmaputra Plain to the high Tibetan Plateau.
Most of the land area of Bhutan is underlain by the Main Himalayan Thrust (MHT), which covers the entire length
of the Himalayan Arc. An indication of fault creeping in some locations is the main reason for the release of
interseismic loading for major earthquakes, and such activities have shown significant microseismicity for Bhutan
(Stevens et al., 2020). Some studies also pointed out seismicity in the eastern fringe of the Himalayan arc can
cause strong to moderate shaking across Bhutan, for example, Berthet et al. (2013).
Historical earthquake catalog (Fig. 1a) indicates that Bhutan has experienced several earthquakes since early
1900 including the 1915 Trashigang (Mw 6.6) and the 1954 Trashiyangtse (Mw 6.4) events. Major destruction
and fatalities were caused by the 2009, Mongar (Mw 6.1) earthquake, which occurred at  11 km east of Bhutan.
The 2011 Sikkim-Nepal earthquake (Mw 6.9) has also caused noticeable damages to building stocks in Bhutan
(Chettri et al., 2021a). The earthquakes in the vicinity of the study area (Phuentsholing) include the 1981
Dagana (Mw 5.1) earthquake, the 1982 Tsimasham (Mw 4.6) earthquake, and the 2003 Haa earthquake (Mw
5.5). Figure 1b shows the very recent 2021 Samtse (Mw 5.1 ) earthquake with intensity scale in the nearby
regions and the Phuentsholing was impacted to an intensity level of IV in Modified Mercalli Intensity (MMI)
scale.

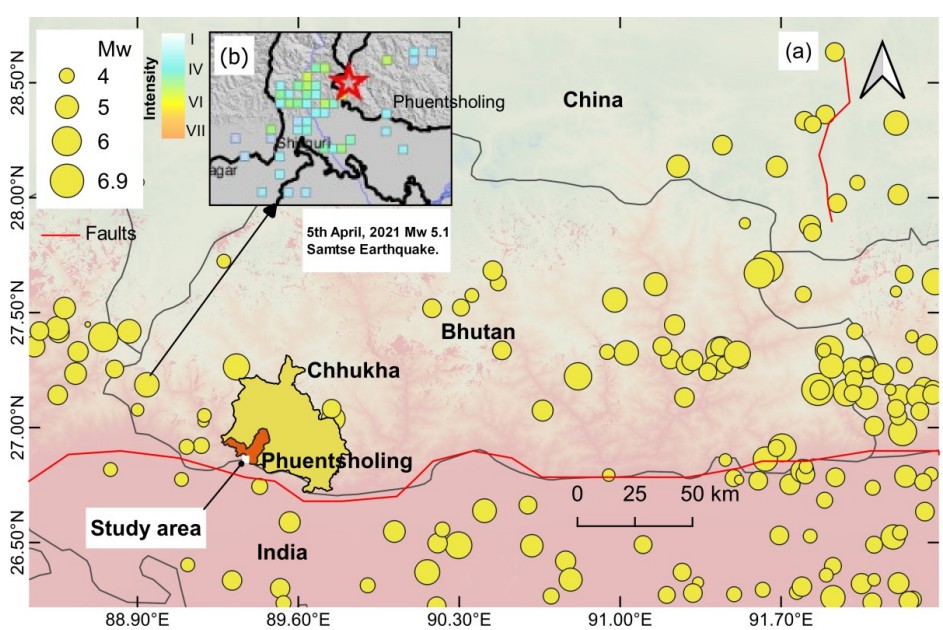

**Figure 1:** Earthquake catalog and scenario, (a) Bhutan Himalaya, (b) Earthquake scenario of 2021 Mw 5.1 Samtse earthquake (modified from: https://www.usgs.gov/).

Figure 2 represents the geological setting of the Phuentsholing and the locations of the available shear waves measurements by MASW profiles are also indicated. Geologically, the Himalayan fringe of Bhutan is one of the most seismically active zones in the world with major shear zones such as Main Himalaya Thrust (MHT), the Main Boundary Thrust (MBT), the Main Central Thrust (MCT), and the South Tibetan Detachment System (STDS) (Long & McQuarrie, 2010). In particular, the current study area falls under the Buxa group of the Lesser Himalayan Zone. The stretches along Zone I comprise rocks belonging to the Phuentsholing formation of the Buxa group, which is characterized by two predominant units such as carbonaceous phyllite and quartzite covered by large deposits of highly weathered colluvium.

A pocket of the Buxa group exists along the left bank of the Toorsa river comprising of deposit of dolomite. Along the proximal bank of Toorsa, the extensive amount of orthogenesis deposition from Daling formation of west-central Bhutan are predominant. The majority of the reference points are picked up from this region and Dhamdhara. Zone II stretches along Rinchending – Pasakha where the structurally fragile crystalline weak film of sub-surface rock of paragneiss, quartzite, schists of Precambrian, orthogneiss, and early Paleozoic ages and Miocene leucogranitic intrusions are present. The belt of differently aged Tethyan sediments (Paleozoic to Eocene) predominated by limestone, shales, and sandstones divides Zone I and II. The region along Shumar formation comprises variegated phyllite and a thin band of greenish-grey quartzite which drifts south every monsoon.





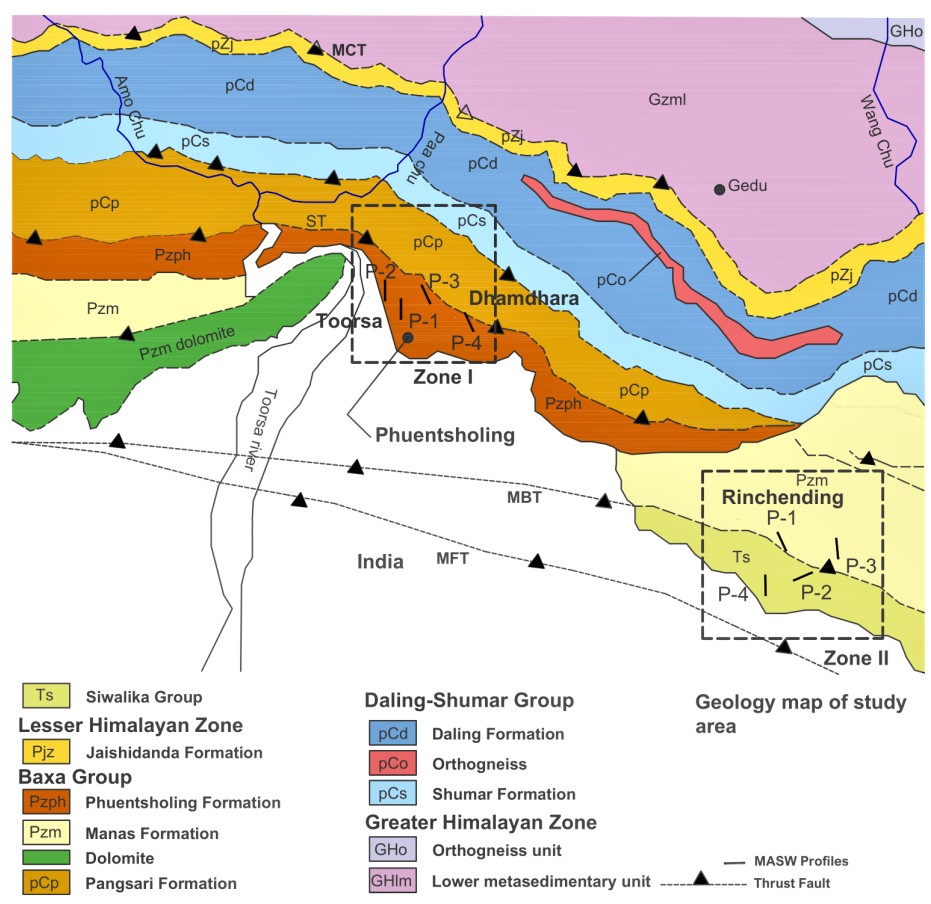

126

**Figure 2:** The geological setting of the study area reproduced from McQuarrie et al. (2013) with the locations of
MASW profiles.

In particular, this study focuses on Phuentsholing Thromde (city) under Chhukha dzongkhag (district) in Bhutan
(Fig. 3). The city is one of the major commercial hubs making the town the gateway to Bhutan for trade with
India. The proposed study area is challenging because of the rapid development activities and expansion of urban
land use to cater to residential, commercial, and industrial transformation besides providing a major trade network
to other districts, e.g., extended Toribari township in the east and Amochu Land Development and Township
Project (ALDTP) in the west. The Phuentsholing city covers an area of 15.6 km$^2$ and is located at 26.86°E and
89.39°N. The city is densely populated with a population of 27658 people, mostly distributed towards the
peripheral international border area with a total of 2263 residential and commercial buildings as of 2020
(http://www.pcc.bt/index.php/). The seismic site characterization includes eight locations in the regions of
Dhamdhara, Toorsa, and Rinchending in Phuentsholing, Bhutan. In this study, the sites are grouped into two main
zones based on the geographical location and the proximity of the survey. The two grouped zones also refer to the
Local Area Plan (LAP) of Phuentsholing. Out of 8 of these LAPs, Dhamdhara and Toorsa (Zone I) fall under the


same region in the western part of the city and Rinchending (Zone II) in the east. A similar classification was also
used by Tempa et al. (Tempa, Chettri, Gurung, et al., 2021). The zones are; Zone I: Dhamdhara I, Dhamdhara II,
Toorsa I and Toorsa II, and Zone II: College of Science and Technology (CST) Football Ground, CST Hostel,
Phajoding, and Monastery area.

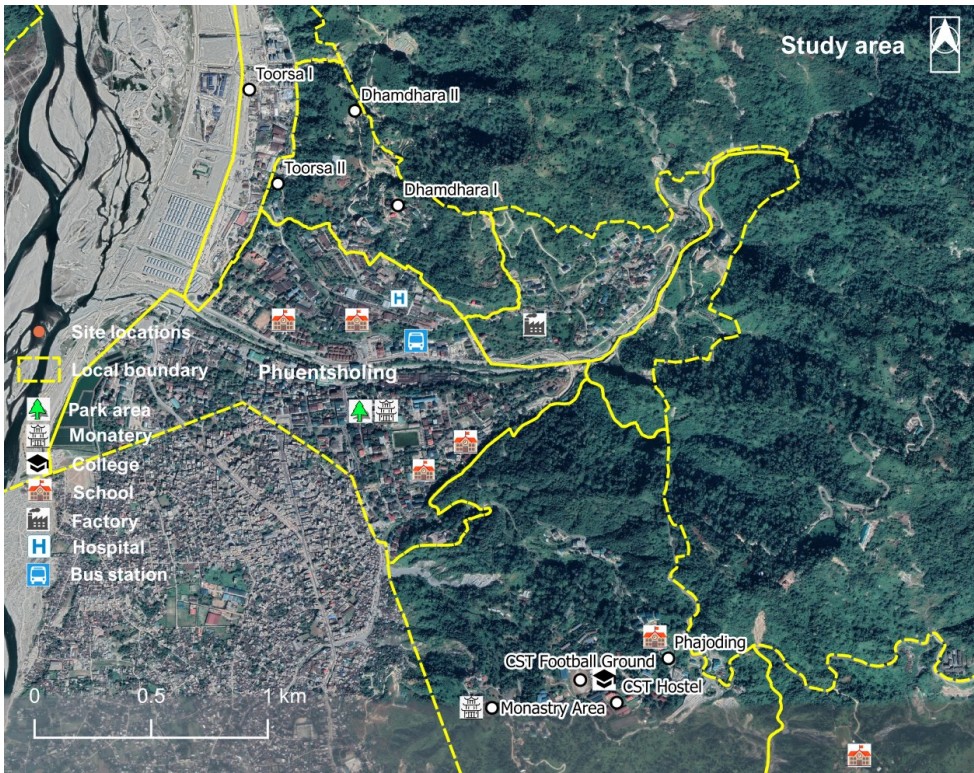


**Figure 3**: Site locations with the distribution of the critical structures in Phuentsholing, Bhutan (modified from
© Google Earth Pro 2021).
**3 Materials and method**
**3.1 Geotecnical site characterization**
From the geotechnical data collected from the Phuentsholing city office, we examined 29 drill log data to
investigate the subsoil profiles and depth of the groundwater table (GWT). The groundwater table in the study
area is shallow and varies between 0.5 m to 16.0 m. The depth of the groundwater table for each of the drilling
logs was mapped in QGIS and interpolated using the inverse distance weighted (IDW) method to capture the
spatial variability of the water table in the study area. Figures 4 and 5 represent the locations of stratigraphic logs
in the study area. Drill log data showed the highest depth of the groundwater table in the Dhamdhara area at a


depth of 12.5 to 16.0 m, while the Rinchending area is underlain at 5 meters, followed by the Toorsa area between
0.5 and 3 m which is located near to the riverbed. The depth of the water table is an essential parameter used for
1D ground response analysis.

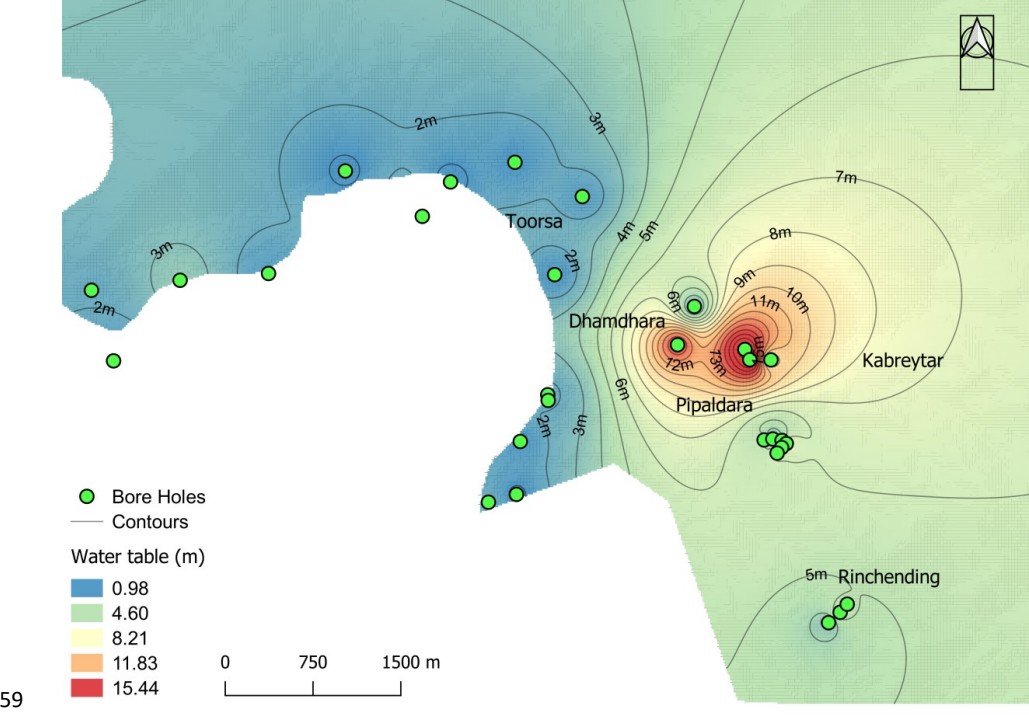


**Figure 4:** Drill log locations and spatial variability of the water table in the study area.
Several soil tests were carried out on undisturbed soil samples taken from boreholes to determine the soil
properties (Table 1). No. of samples shown in Table 1 in respective zones represents the total number of samples
collected from all the drill logs at different stratigraphic depths, which includes disturbed and split spoon samples.
For each sample, the corresponding tests stated in Table 1 were conducted. All the tests were conducted as per the
Indian standard code. Laboratory tests include Atterberg limits, sieve analysis, and direct shear test to obtain soil
consistency, grain size distribution, and shear strength parameters. Field tests such as the standard penetration test
(SPT) and the core cutter test were conducted to determine the penetration resistance ($N$-value) and the field
density respectively.



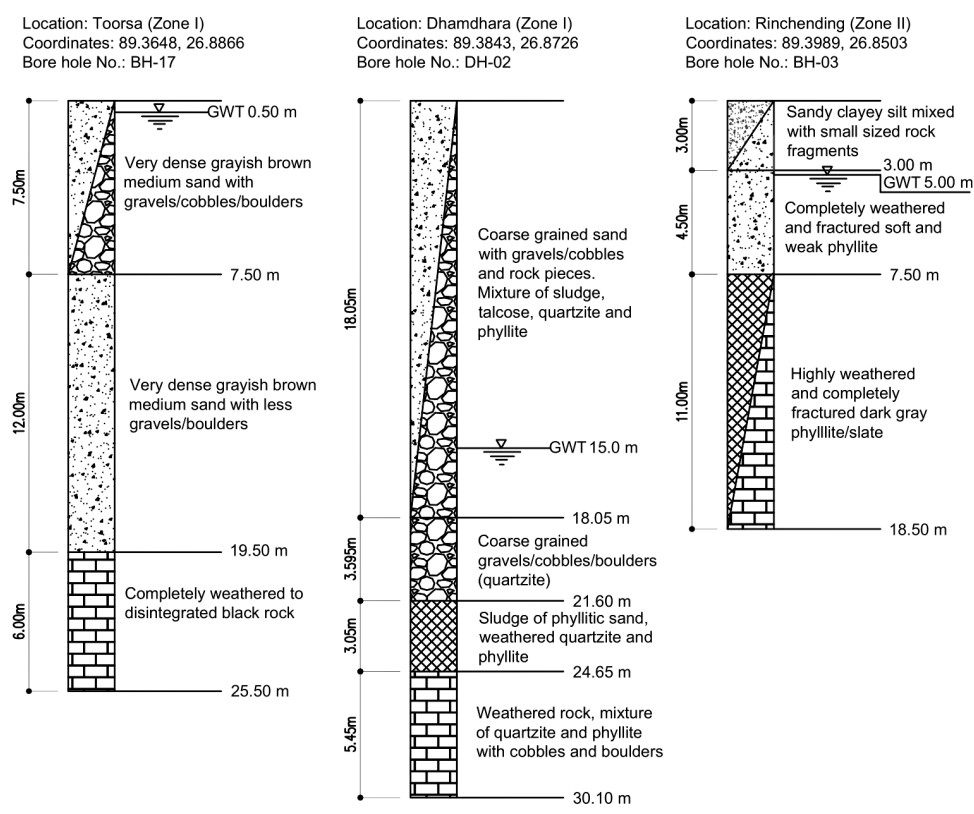


**Figure 5:** Typical borehole stratigraphies in Toorsa and Dhamdhara (Zone I) and Rinchingding (Zone II).

As shown in the stratigraphic logs reported in Fig. 5, the shallow soils are predominantly mixed coarse-
grained soils mainly made of sand with a high proportion of gravel and a good proportion of fines and can be
classified as SG-SM and SP (NP) (Tempa and Chettri, 2021; Tempa, Chettri, Sarkar, et al., 2021). The soil
classification of the Phuentsholing area carried out by sieve analysis highlighted that the majority of soils consist
of 22.74% gravel, 74.89% sand, and 2.37% of the silt and clay fractions. The sieve analysis results for the
respective zones are shown in Fig. 6. The soils in Toorsa are non-plastic, as coarser grained soils dominate the
particle distribution, while the soils in Rinchending and Dhamdhara had a low plasticity with a plasticity index
(PI) of 6.5 and 10, respectively. The bulk density is 1.8 g/cm$^3$ in Toorsa, 1.64 g/cm$^3$ in Dhamdhara, and 1.33 g/cm$^3$
in Rinchending. The shear strength parameter cohesion (c) ranges between 0-0.18 kg/cm$^2$, while the angle of
internal friction (φ) in the study area is up to 35 °.




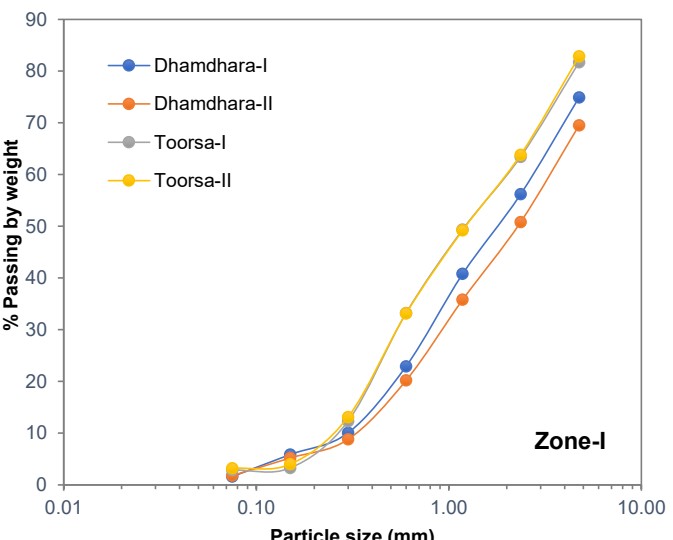


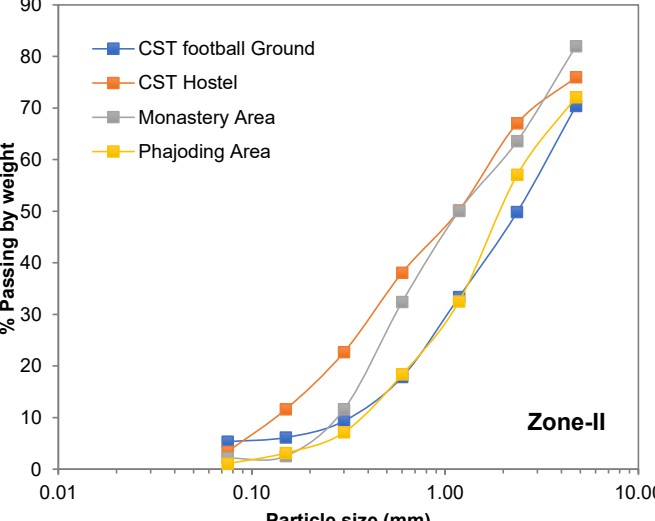


**Figure 6:** Results of sieve analysis showing grain size distribution curves.

**Table 1.** Average soil parameters in the study area.

| Location | Testing methods | Soil parameters | No. of samples | Reference |
|---|---|---|---|---|
| Toorsa (Zone I) | Atterberg's limit | Non-plastic | 86 | IS: 2720 (Part 5)-1995 |
| | Core cutter | Bulk density, $\gamma_t$ = 1.8 g/cc | | IS:2720 (Part 29)-1975 |





| | | Dry density, $\gamma_d$ = 1.64 g/cc | | |
| | Direct shear | c = 0 <br> ϕ = 35˚ | | IS: 2720 (Part 13)—1997 |
| | SPT | $N$-value = 25 to 50 | | IS: 2131–1981 |
| Dhamdhara (Zone I) | -do- | Low plasticity (PI = 6.5) <br> Bulk density, $\gamma_t$ = 1.64 g/cc <br> Dry density, $\gamma_d$ = 1.51 g/cc <br> c = 0.073 kg/cm² <br> ϕ = 31.44˚ <br> $N$-value = 19 to 37 | 28 | |
| Rinchending (Zone II) | -do- | Low plasticity (PI = 10) <br> Bulk density, $\gamma_t$ = 1.33 g/cc <br> Dry density, $\gamma_d$ = 1.70 g/cc <br> c = 0.18 kg/cm² <br> ϕ = 20-30˚ <br> $N$-value = 21 to >100 | 26 | |


186         In most cases, the shear wave velocity of the site does not extend over a depth of more than 30 m depth
and $V_{S,30}$ is widely used in seismic site characterization. Shear wave velocity profiles from eight locations in the
study area based on the Multispectral Analysis of Surface Waves (MASW) (Fig. 7) and empirical correlation
developed by Tempa et. al. (Tempa, Chettri, Gurung, et al., 2021) are used to carry out ground response analysis.
According to the shear wave velocity profile, the engineering bedrock ($V_s$ > 800 m/s) lies at a depth ranging from
150 m to 400 m (e.g., Dhamdhara I in Zone I and Phajoding in Zone II), as shown in Fig. 8. According to the
parametric analysis carried out by (Tempa et al., 2020) in the study area, the site condition is classified to ground
type B in conjunction to Euro Code EC-08 and National Earthquake Hazards Reduction Program (NEHRP) with
the majority of shear velocity ($V_{s,30}$) ranging between 380–470 m/s, except in the Phajoding which has a shear
velocity of 584.76 m/s (Table 2). The $V_{s,30}$ can be estimated with the following Equation 1. The higher values of
the shear wave velocities of the soil layers in all zones indicate a low liquefaction potential in the study area. The
scope of the current study does not include the assessment of liquefaction susceptibility.
$$V_{s,30} = 30 \Big/ \sum_{n=1}^{N}\left(\frac{h_i}{V_i}\right), \text{ m/s} \tag{1}$$




**Figure 7:** Typical 2D shear wave profiles; (a) Dhamdhara I: Zone I, (b) CST hostel: Zone II (Norwegian Geotechnical Institute (NGI), 2009).

**Table 2**. Site classification as per Euro Code EC08

| Zones | Sites | $V_{s,30}$ (m/s) | Ground Type |
|---|---|---|---|
| I | Dhamdhara I | 386.43 | B |




| | | | |
|---|---|---|---|
| | Dhamdhara II | 435.92 | B |
| | Toorsa I | 439.54 | B |
| | Toorsa II | 464.30 | B |
| | CST football ground | 426.76 | B |
| II | CST hostel | 426.61 | B |
| | Monastery area | 446.20 | B |
| | Phajoding | 584.76 | B |
| All | Bedrock | >800 | A |

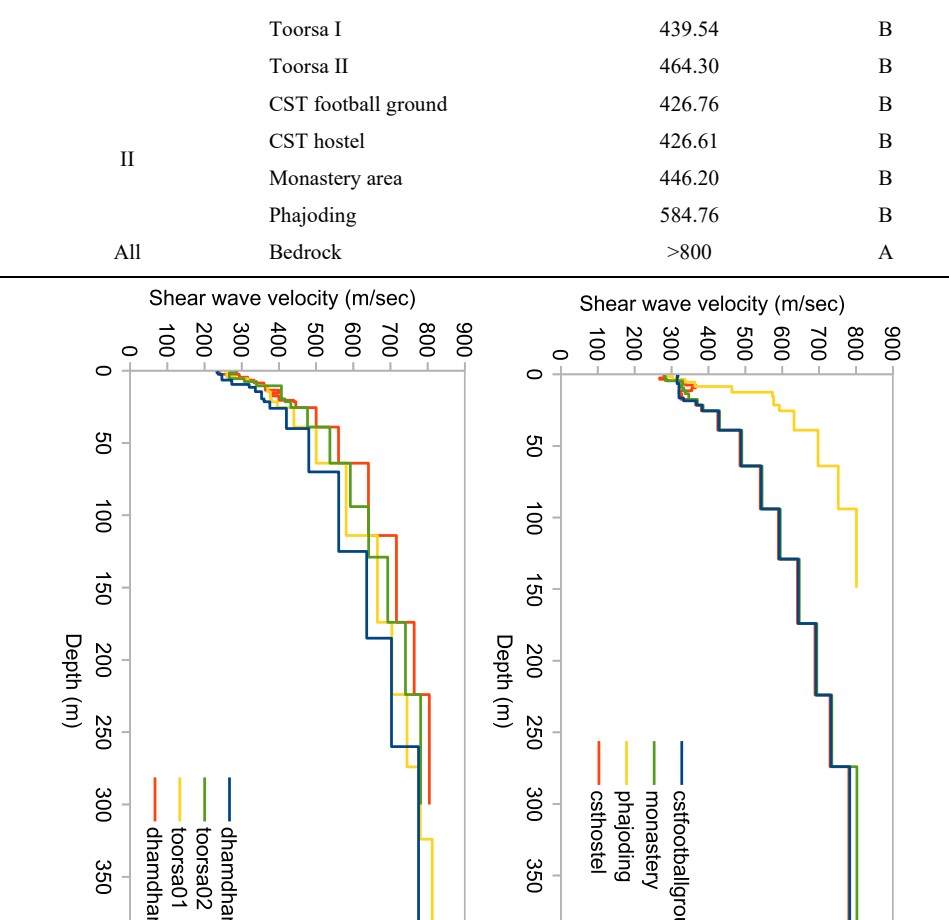

**Figure 8:** Shear wave velocity profile of study locations in Phuentsholing, Bhutan.

To further supplement the requirements for the seismic demand and damage risk, it is essential to take into account the subsurface conditions associated with the earthquake energy, which amplify or abbreviate the ground motion responses (Kramer, 1996). Dynamic properties of soils are influenced by shear modulus and damping and are defined by the respective degradation models. Figure 9 represents the dynamic soil model for sand used in this study. Degradation models are well established by many researchers for different types of soils, which influence the response at low strain levels, see e.g., (Seed and Idriss, 1970; Vucetic and Dobry, 1991; Darendeli, 2001; Dobry and Vucetic, 1982; Seed et al., 1986).

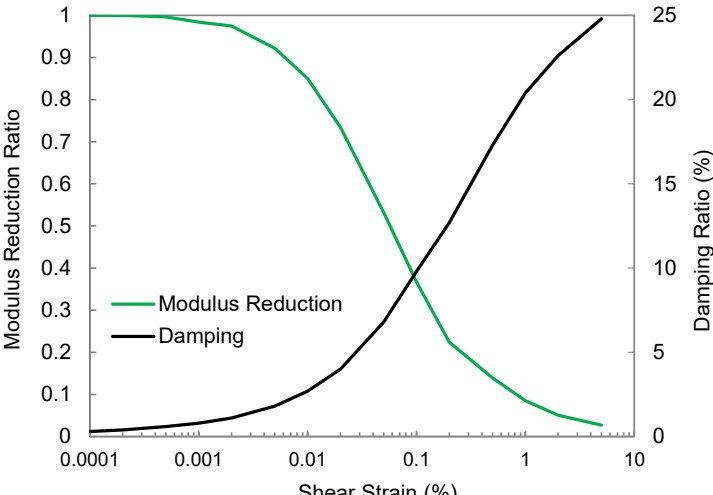


**Figure 9:** Average modulus reduction ratio and damping ratio adopted for sand (Seed & Idriss, 1970).

A damped linear elastic model of the soil system is used. The nonlinearity of the soil for which the shear
modulus is strain-dependent, ProShake performs an iterative process on the linear model until both moduli and
damping ratios are compatible to average strains and convergence is achieved at the final iteration (Shafiee et al.,
2011; Puri et al., 2018). The nonlinear and hysteretic stress-strain behavior of soils is approximated during cyclic
loadings as a function of $G_{sec}$ and $G_{max}$. This predetermined estimation of $G_{sec}$ or G and $G_{max}$ is attributed by unit
weight or bulk density, $\rho$, and shear wave velocity, $V_s$ ( $G_{max} = \rho V_s^2$ ). Similarly, damping ratios are predicted as a
function of $G_{sec}$ or $G$ values. This estimation is achieved with the iterative procedure in Proshake 2.0 program
(EduPro Civil Systems Inc., 2017).
**3.2 Selection of input motion**
The study of the site response of an area requires a good subsoil characterization and a careful selection of
accelerograms for defining the input motion. In Bhutan, records of acceleration time histories are very scarce.
Stochastically simulated ground motions are also commonly used worldwide in site response studies. Both the
simulated and the recorded ground motions can, however, be used for site-specific response studies (Ansal &
Tönük, 2007). In absence of a national seismic code, Bhutan is assumed to fall under Indian seismic zone IV
V with PGA of 0.24 g and 0.36 g respectively for Maximum Considered Earthquake (MCE) (IS:1893, 2002), and
in many instances, PGA of 0.36 g is applied uniformly across the entire country (Stevens et al., 2020). For the
two zones mentioned, the PGA for earthquakes with a return period of 475 years is expected to be half of MCE,
i.e., 0.12 g and 0.18 g. From the global seismic map (GSHAP), the PGA depicted are in the range 0.20 g to 0.28g,
with an increasing pattern to the east of the country (Tempa, Chettri, Gurung, et al., 2021). The discrepancies in
such agreements without much conformity lead to a question about how the earthquake scenario is differently


distributed at the regional level at the current juncture. In this study, such observations have been instrumental in
the selection of six historical global earthquakes as input motion having an intensity of PGA in the range of 0.067
g to 0.422 g considering the least and the highest range of possible earthquake scenarios (Table 3).
Most commonly, for engineering purposes, two characteristics of earthquake motion, i.e., amplitude and
frequency content of the motion at bedrock level are of primary importance (Kirtas et al., 2015; Kramer, 1996).
The acceleration time histories used for 1D ground response analysis are shown in Fig. 10 in ascending order of
PGA using ProShake 2.0 computer program. In the ProShake 2.0 program, input motion and soil profile are
denoted as "M" and "P" respectively and are used in the following sections. To understand the strong ground
motion characteristics, we plotted Fourier amplitude versus period in the frequency domain (or period), which
represents Fourier amplitude spectra of input motions, as shown in Fig. 11. The effect of local soils is indicative
at a much higher frequency range in all the investigated sites.
**Table 3.** Historical earthquakes considered as input motion.

| Event | Station | Year | $M_w$ | PGA (g) |
|---|---|---|---|---|
| Loma Prieta/Santa Cruz Mountains | Yerba Buena Island, CA – US Coast Guard | 1989 | 6.9 | 0.067 |
| Loma Prieta | Diamond Heights | 1989 | 6.9 | 0.113 |
| Taft Kern County | Taft | 1952 | 7.5 | 0.185 |
| Northridge | Topanga Fire Station | 1994 | 6.7 | 0.329 |
| El Centro | Imperial Valley Irrigation District | 1940 | 6.9 | 0.344 |
| Petrolia | Cape Mendocino | 1992 | 6.6 | 0.422 |







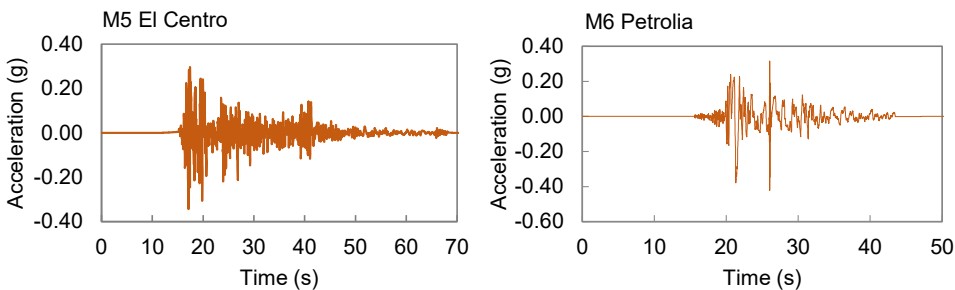

**Figure 10:** Acceleration time histories considered to account for the effect of variability of input ground motions to the ground response.

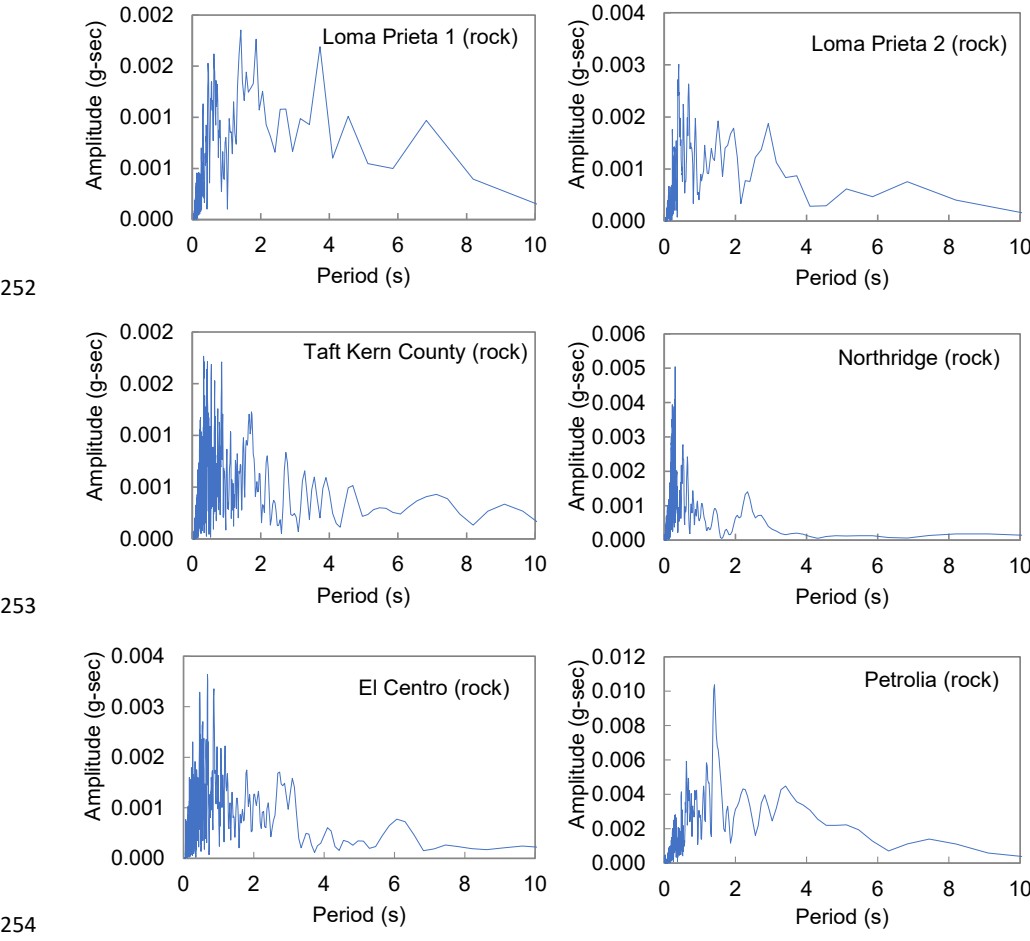

**Figure 11:** Amplitude characteristics of input ground motions.


### 3.3 1D ground response analysis

A 1D equivalent linear analysis was performed at eight sites in Phuentsholing, Bhutan to estimate local site effects with the ProShake 2.0 program. In this study, six strong motion records were used to replicate low, medium, and high earthquake accelerations. The ProShake 2.0 program offers the flexibility to input ground motions and soil profiles and is handy to estimate the outcrop responses to input ground shaking. The enhanced shear wave velocity profiles to the depth of the engineering bedrock (150 m and 400 m) of eight locations based on initial MASW profiles of ~ 22.2 m depth is well established in the study conducted by Tempa et. al. (Tempa, Chettri, Gurung, et al., 2021). These deep shear wave profiles used are a complementary input parameter in the current study which takes into account the effects of depth of embedment of varying visco-elastic soil strata underlain with the predicted engineering bedrock.

The 1D ground response analysis takes into account the scope of wave propagation from the bedrock outcrop through the visco-elastic layered soil deposit and the ground surface motion parameters are estimated. From geotechnical engineering propestive, these parameters are usually referred to as seismic hazard parameters (Kramer, 1996). The complex response method is solved by the equation of motion in the frequency domain. The soil nonlinear response is estimated by an iterative, quasi-linear procedure in which successive linear analyses are performed, with the soil shear modulus and damping ratio are updated based on the shear strain level obtained in the previous analysis. Iterations continue until the strain-compatible modulus and damping converge.

## 4 Results and discussion

### 4.1 Variation in ground motion parameters

The PGA represents the intensity of earthquake shaking and it is indispensable to demonstrate the destructive nature of the particular earthquake. The PGA at the surface in two typical sites of two regions in Phuentsholing shows approximately 0.1g to 0.15g for low-intensity earthquakes, 0.23g to ~0.38g for moderate-intensity earthquakes, and more than 0.43g for high-intensity earthquakes such as the 1992 Petrolia earthquake (Fig. 12).

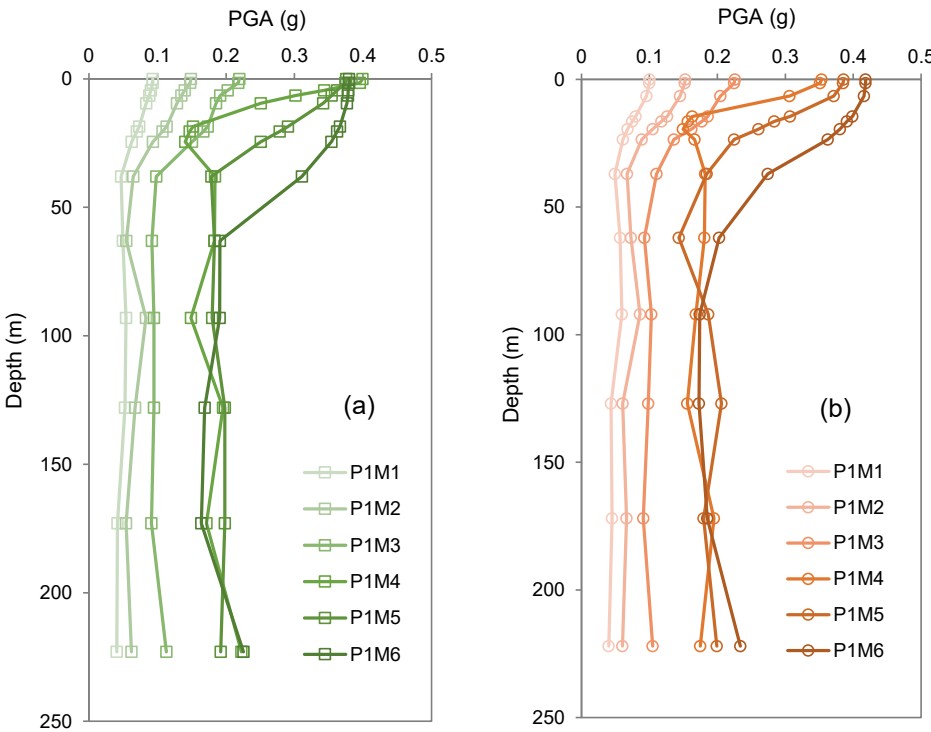

**Figure 12:** The typical profile of peak ground acceleration (PGA), (a) Soil profile P1 of Toorsa II in Zone I, and
(b) Soil profile P1 of CST Football Ground in Zone II.

Spectral acceleration (SA) defines the maximum acceleration to a response of a damped, harmonic
oscillator in one physical dimension, usually a system with a single degree of freedom (SDOF), and is measured
in "g", the acceleration due to gravity at various frequencies. In addition, response parameters can be defined and
characterized based on the amplitude parameters of the ground motion and the severity of the ground motion to
on the respective structure. This in turn is a function of the amplitude or intensity, the frequency content, and the
duration of the ground motion (Bradley, 2011). Natural periods or frequency domain parameters are well related
to the seismic behavior of structures and indirectly reflect the ground motion characteristics (Zafarani et al., 2020).
This relationship can be understood by a response spectrum plot, as shown in Figs. 13 and 14, which depicts the
surface and bedrock spectra.

Taking into account the local site conditions in the study area, the response to various input ground
motions indicates a higher spectral acceleration of the soil profile in the period range from 0.3 s to 3.0 s with
approximately 0.14 g to 1.62 g peak spectral acceleration. Buildings 3 m to 30 m tall usually fall into this spectrum.
In the city of Phuentsholing, buildings with 2-8 storeys can show higher values of hazard responses due to the
variability of the earthquake shaking intensity and soil condition examined. Both study areas show a similar
tendency of ground responses as expected.




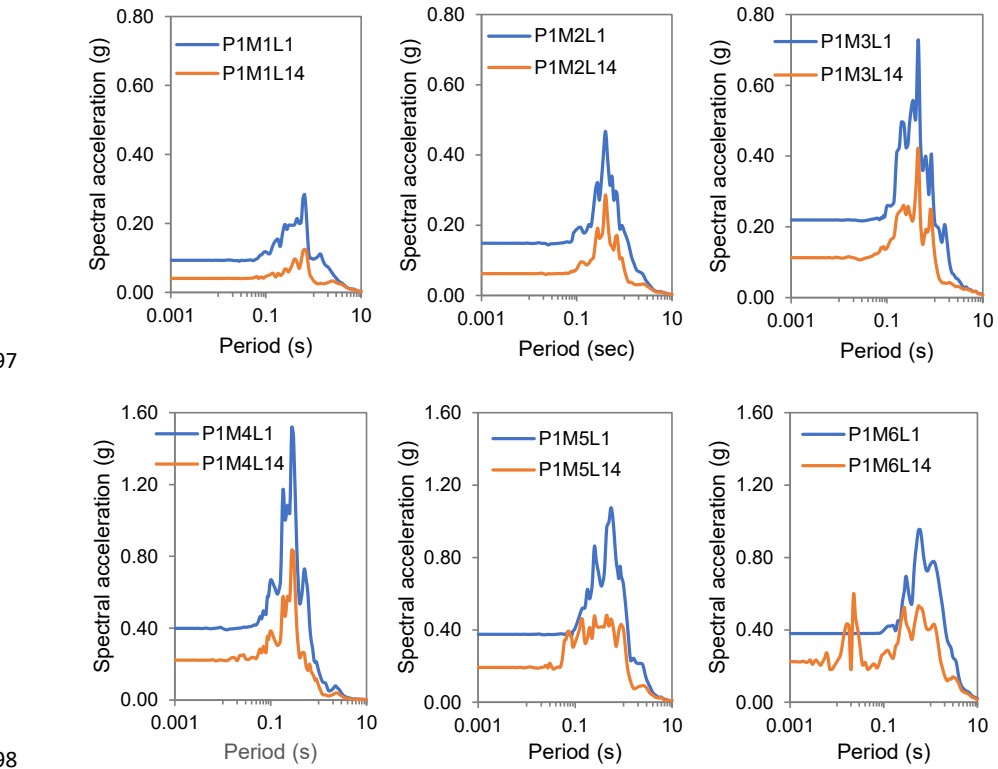

297

298

**Figure 13:** Spectral acceleration for each soil profile corresponding to various input ground motions in Zone I:
(P) Soil profiles and (M) Input ground motions – M1: Yerba Buena Island/Loma Prieta, M2: Loma Prieta, M3:
Taft Kern County, M4: Northridge, M5: El Centro and M6: Petrolia. Soil profile number indicates: P1: Toorsa II,
P2: Toorsa I, P3: Dhamdhara II, P4: Dhamdhara I, L1: Surface, and L14: Bedrock.

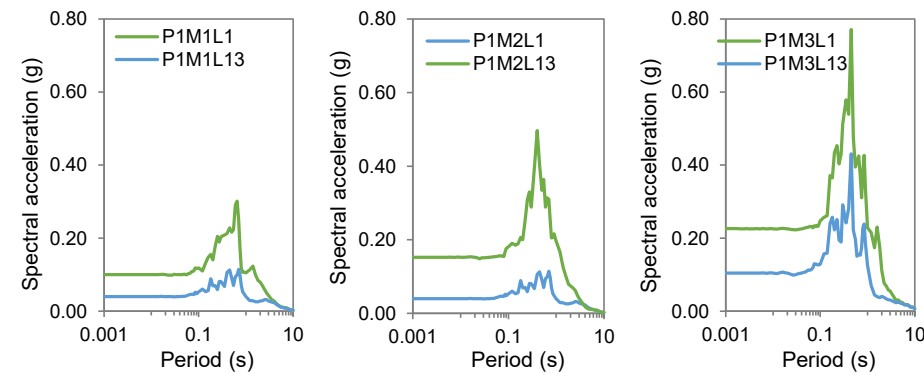

303


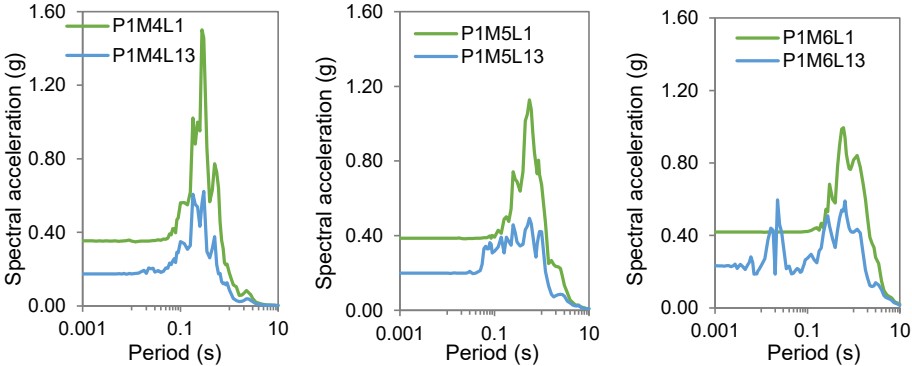

304

**Figure 14:** Spectral acceleration for each soil profile corresponding to various input ground motions in Zone II:
(P) Soil profiles and (M) Input ground motions – M1: Yerba Buena Island/Loma Prieta, M2: Loma Prieta, M3:
Taft Kern County, M4: Northridge, M5: El Centro, and M6 Petrolia. Profile number indicates P1: CST Football
Ground, P2: CST Hostel, P3: Phajoding, P4: Monastery area, L1: Surface, and L14: Bedrock.

A key parameter for taking into account the modification of seismic waves due to local site conditions is
usually illustrated by the amplification factor (Bhutani and Naval, 2020). Figures 15 and 16 show the results of
typical amplification factors at two locations in the study area. Amplification is known as an increase in earthquake
ground motion intensity, which can be greater than expected for solid ground or rock. This is usually the case
when the predominant period of the site soils underlaid with thick, soft soil deposits or the soil coincides with the
predominant period of the earthquake motion. The ratio of the spectral acceleration of the soil surface to the
bedrock spectral acceleration provides the amplification factor that is determined using Eq. 2. These results show
that the characteristics of input ground motion arehighly correlated with the soil conditions of the site.
$$Amp(T) = \frac{SA_{Soil}(T)}{SA_{Rock}(T)}$$     (2)
From the results of the ground response analysis, the amplification factors in the study areas can be
broadly classified into three categories as low, medium, and high ranges. The amplification factors range from 0.7
to 2.7, 0.6 to 2.6, 0.75 to 2.5, and 0.7 to 3.2 for Toorsa II, Dhamdhara I, CST football ground, and Phajoding
respectively for 0.01 s to 0.1 s natural period. In the natural period range from 0.1 to 1.0 s, the amplification factors
are in the range from 1.1 to 3.6, 0.7 to 4.2, 1.0 to 3.7, and 1.2 to 5.2 for Toorsa II, Dhamdhara I, CST football
ground, and Phajoding respectively. In the high natural period range, the amplification factors are 5.0, 6.2, and
5.8 for Toorsa II, Dhamdhara I, and CST football ground respectively. In the Phajoding, however, the significance
of amplification is ~ 1.7 due to a much stiffer soil deposit ($V_{s,30}$ = 584.76 m/s) and shallow engineering bedrock at
150 m.

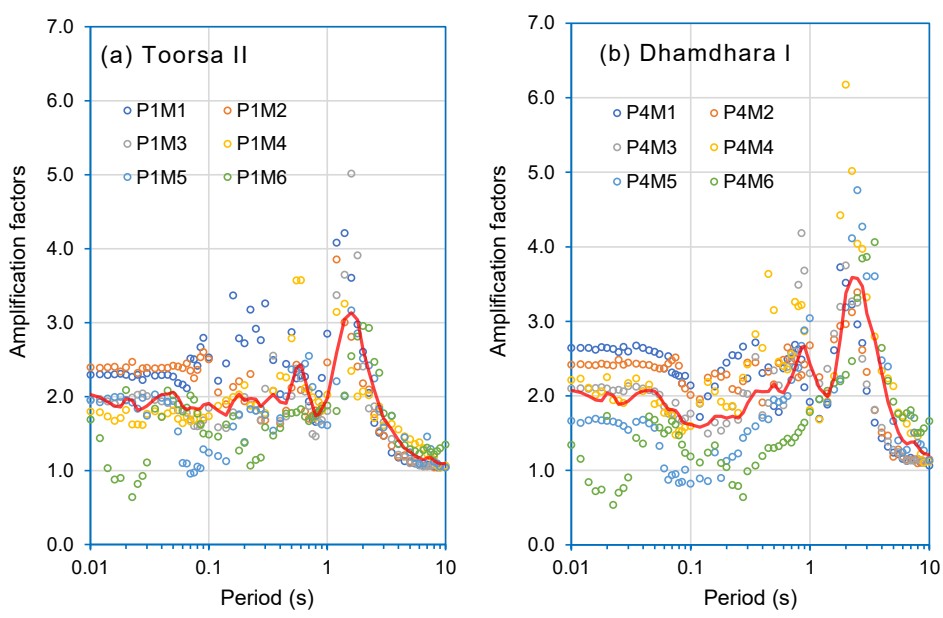

**Figure 15:** Typical amplification factors for various earthquakes at (a) Soil profile P1 Toorsa II in Zone I, (b) Soil profile P4 Dhamdhara I in Zone I; Input ground motions – M1 Yerba Buena Island/Loma Prieta, M2 Loma Prieta, M3 Taft Kern County, M4 Northridge, M5 El Centro, and M6 Petrolia.

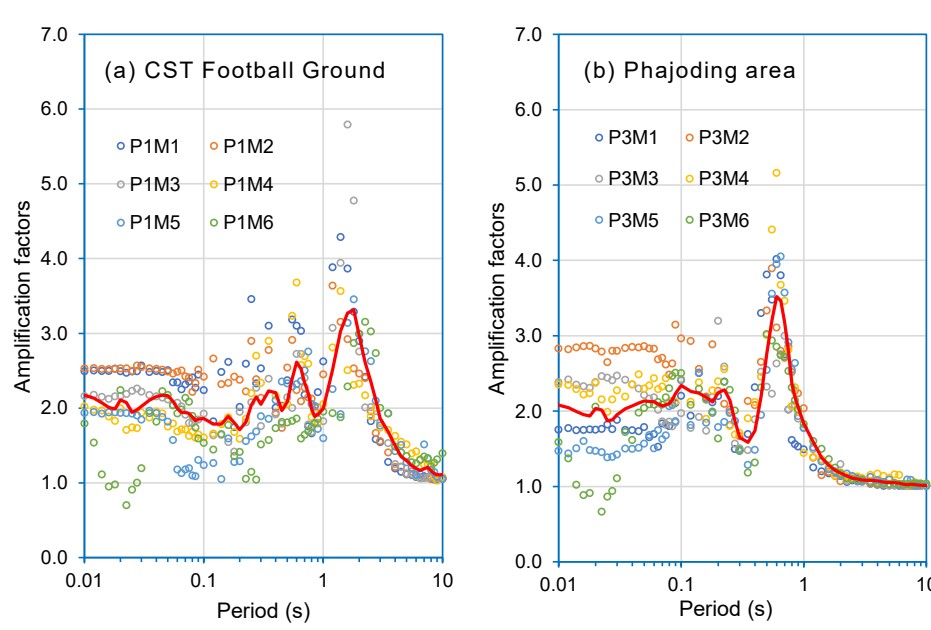


**Figure 16:** Typical amplification factors for various earthquakes at (a) Soil profile P1 CST Football Ground in
Zone II, (b) Soil profile P3 Phajoding in Zone I; Input ground motions – M1 Yerba Buena Island/Loma Prieta,
M2 Loma Prieta, M3 Taft Kern County, M4 Northridge, M5 El Centro, and M6 Petrolia.
**4.2 Sensitivity of input motion**
As the various earthquake ground motion propagate through different soil profiles, the ground surface motion
response is modified in the ascending order of the PGA of input motion. Since all the sites fall under the type B
site, the trend of the variation in the ground motion to surface are very similar, so the average values may be
decisive for improving the realization of the scenario-based seismic risk in the study area. Ground response
parameters such as PGA and response spectrum intensity including Aries intensity show linear variations for
aggregated values with increasing intensity of earthquake shaking corresponding to a particular soil profile.
However, the predominant period does not show a linear correlation with the characteristics of strong ground
motion. These results were mainly observed due to change in characteristics of seismic waves propagating through
different stratified soil deposits before the strong ground motion reach the ground surface. The plot of sensitivity
of various input motions on amplitude parameters to different local soils in two study zones is shown in Figs. 17
and 18.
Within the set of predominant natural periods corresponding to each input motion, the standard deviation
is lower compared to the data set of the response spectrum of the soil column, which indicates a higher strength
of soil response to the SDOF system, as presented in Figs. 17d and 18d. The non-linearity of soils often shows a
significant scatter in spectral acceleration at higher and lower periods, and therefore the practical reliability of the
result is that it requires more analysis with larger sets of input motions will be required to predict the mean (or
median) response with some level of confidence (Kramer et al., 2012). Conceptually, the standard deviation is a
measure for the amount of variation of a set of data with respect to the mean value of the dataset. Tables 4 and 5
summarize the statistical results of seismic response parameters indicating the sensitivity of various earthquake
inputs at local sites in Zone I and Zone II with additional ground response parameters provided in Tables A1 and
A2. The sensitivity of the output results is shown in Figs. 15 and 16 with examples from two site locations. Ground
motion M4 (Northridge) and M5 (El Centro) show a slight spread of ground response compared to others ground
motions. Such an effect is attributed to the high frequency content at very low natural periods of the ground motion
which is indicative in the Fourier amplitude representation, as shown in Fig. 8. Based on this sensitivity analysis,
the correlation analysis is carried out in the subsequent section and ground response parameters are projected in
order to provide probable seismic hazard scenarios for input levels.
**Table 4.** Statistics of averaged ground response parameters in Zone I for all four soil profiles and six input
ground motion.

| | PGA (g) | Aries intensity (m/sec) | Response spectrum intensity (g²) | Predominant period (sec) | Mean frequency (Hz) |
|---|---|---|---|---|---|
| Mean | 0.270 | 1.073 | 2.996 | 0.818 | 3.527 |




| | | | | | |
|---|---|---|---|---|---|
| Median | 0.238 | 0.630 | 2.450 | 0.689 | 3.319 |
| Standard deviation | 0.121 | 0.765 | 2.013 | 0.468 | 1.097 |
| 84th percentile | 0.407 | 2.215 | 4.541 | 1.251 | 4.824 |
| 16th percentile | 0.139 | 0.179 | 1.322 | 0.379 | 2.283 |


**Table 5.** Statistical relationship of averaged ground motion parameters in Zone II for all four soil profiles and
six input ground motion.

| | PGA (g) | Arias intensity (m/s) | Response spectrum intensity (g²) | Predominant period (s) | Mean frequency (Hz) |
|---|---|---|---|---|---|
| Mean | 0.271 | 1.079 | 2.985 | 0.812 | 3.814 |
| Median | 0.237 | 0.622 | 2.417 | 0.684 | 3.538 |
| Standard deviation | 0.126 | 0.794 | 2.066 | 0.453 | 1.382 |
| 84th percentile | 0.411 | 2.226 | 4.541 | 1.243 | 5.330 |
| 16th percentile | 0.136 | 0.174 | 1.287 | 0.377 | 2.349 |


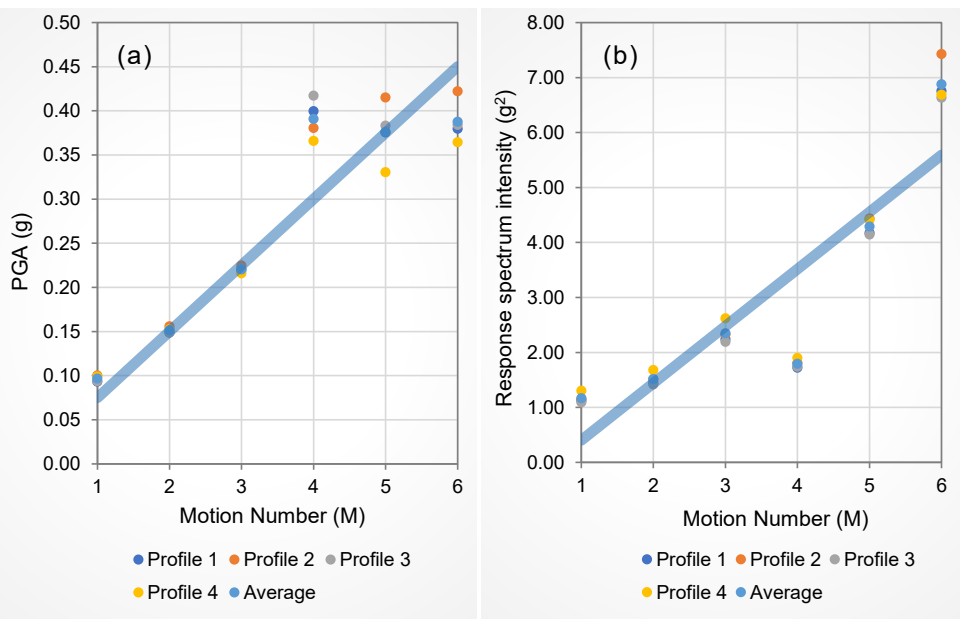


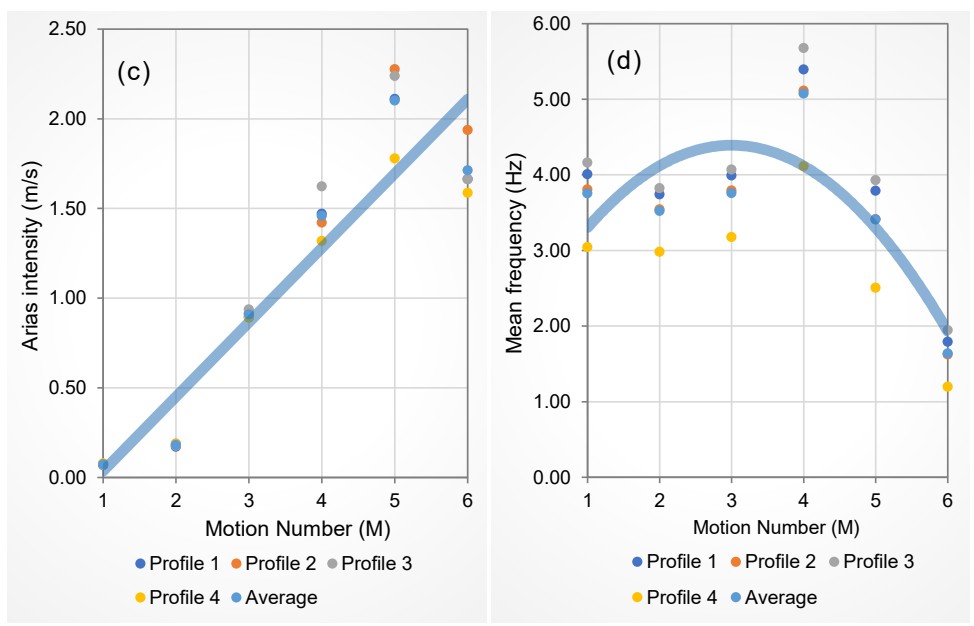

**Figure 17:** Sensitivity of input ground motion in Zone I. Motion number: M1 Yerba Buena Island, M2 Loma Prieta, M3 Taft Kern County, M4 Northridge, M5 El Centro, M6 Petrolia; Soil profile number: P1 Toorsa II, P2 Toorsa 1, P3 Dhamdhara II and P4 Dhamdhara I.

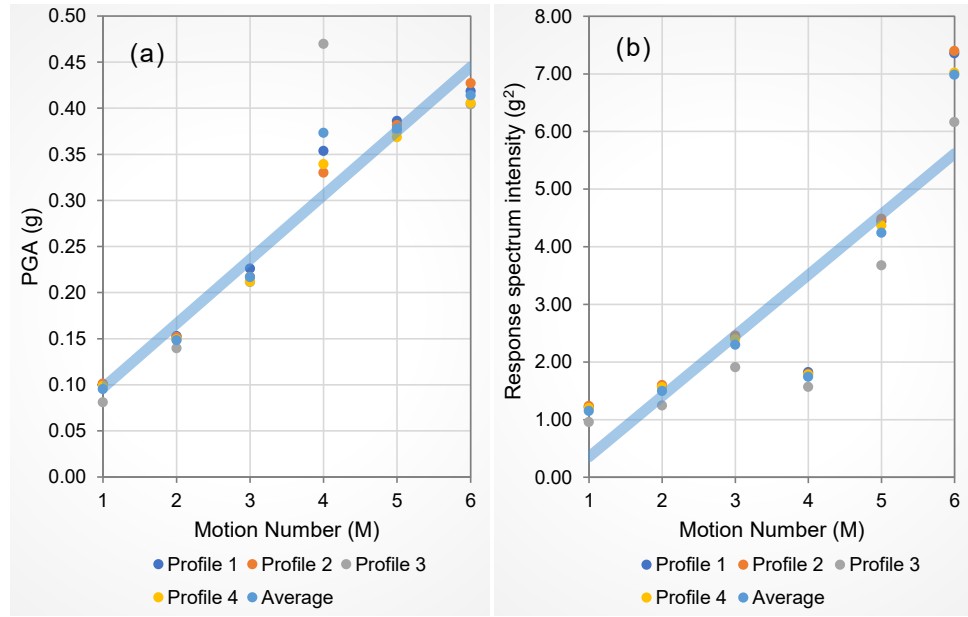
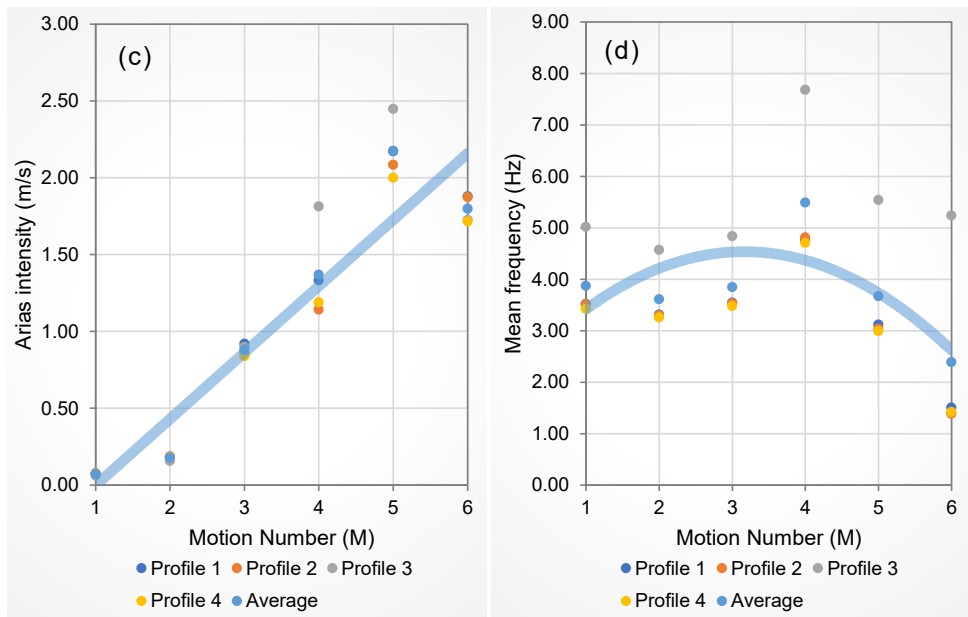

**Figure 18:** Sensitivity of input ground motion in Zone II. Motion number: M1 Yerba Buena Island, M2 Loma Prieta, M3 Taft Kern County, M4 Northridge, M5 El Centro, M6 Petrolia; Profile number: P1 = CST Football Ground, P2 = CST Hostel, P3 = Phajoding, and P4 = Monastery area.

The sensitivity examination of input ground motion to local soils in eight sites indicates the variability of PGA. In the context of sensitivity diagnosis, as shown in Figs. 15 and 16, a set of PGA values of M4 Northridge is mapped to deduce spatial variability of PGA values in two study zones, as shown in Fig. 19. The ambiguity of effects owing to local soils has been noted with higher intensity earthquakes, in particular the Northridge earthquake. The variability of PGA in Zone II is higher compared to Zone I resulting in the range 0.33 g to 0.47 g. The resulting interplay of strong ground motion with local soil conditions primarily highlights the importance of the current study on the significance of input motion characterization. According to the current sensitivity diagnosis, an approach of testing a hypothetical framework could also help narrow the desired earthquake scenarios using the global or regional earthquakes.




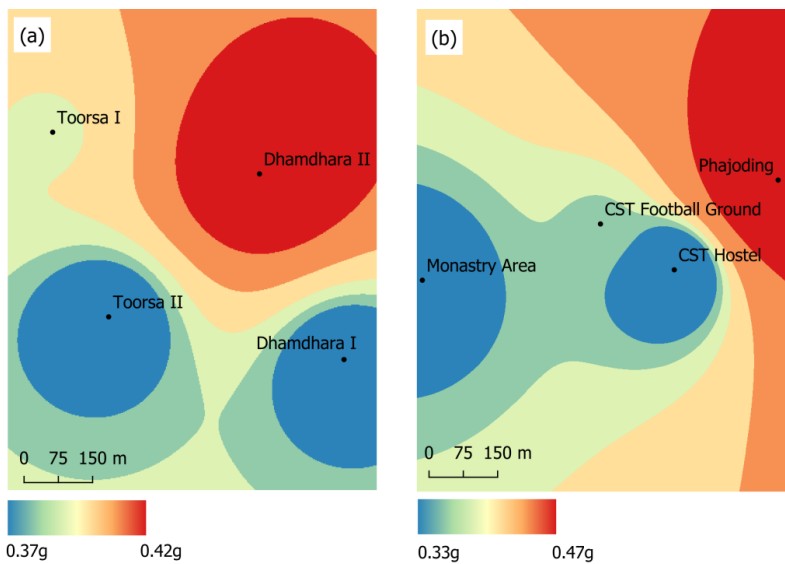

**Figure 19:** PGA map of input motion M4 Northridge earthquake, (a) Toorsa and Dhamdhara in Zone I, (b) Rinchending in Zone II.

### 4.3 Correlation analysis

Statistical correlations are fitted between the ground motion parameters such as Peak Ground Acceleration (PGA), Peak Ground Velocity (PGV), Peak Ground Displacement (PGD), and Spectral Acceleration (SA) to determine the interplay between the effects of strong ground motion and the local soil conditions. According to the results shown in Figs. 20 and 21, the overall trend of seismic responses is in an increasing order of magnitude of earthquake tremors of stronger ground motion. As expected, the 1992 Petrolia earthquake with 0.422 g PGA (Mw = 6.6) led to the highest response; However, the 1994 Northridge earthquake with a PGA of 0.329 g (Mw = 6.7) shows greater variability in amplitude parameters such as peak displacement. From the perspectives of site responses in both the zones in the investigation area, the scatter diagram of the spectral acceleration (period or frequency domain) is widely scattered, which indicates uncertainty in the soil response characteristics that could largely impact the building coinciding with the fundamental natural frequency.




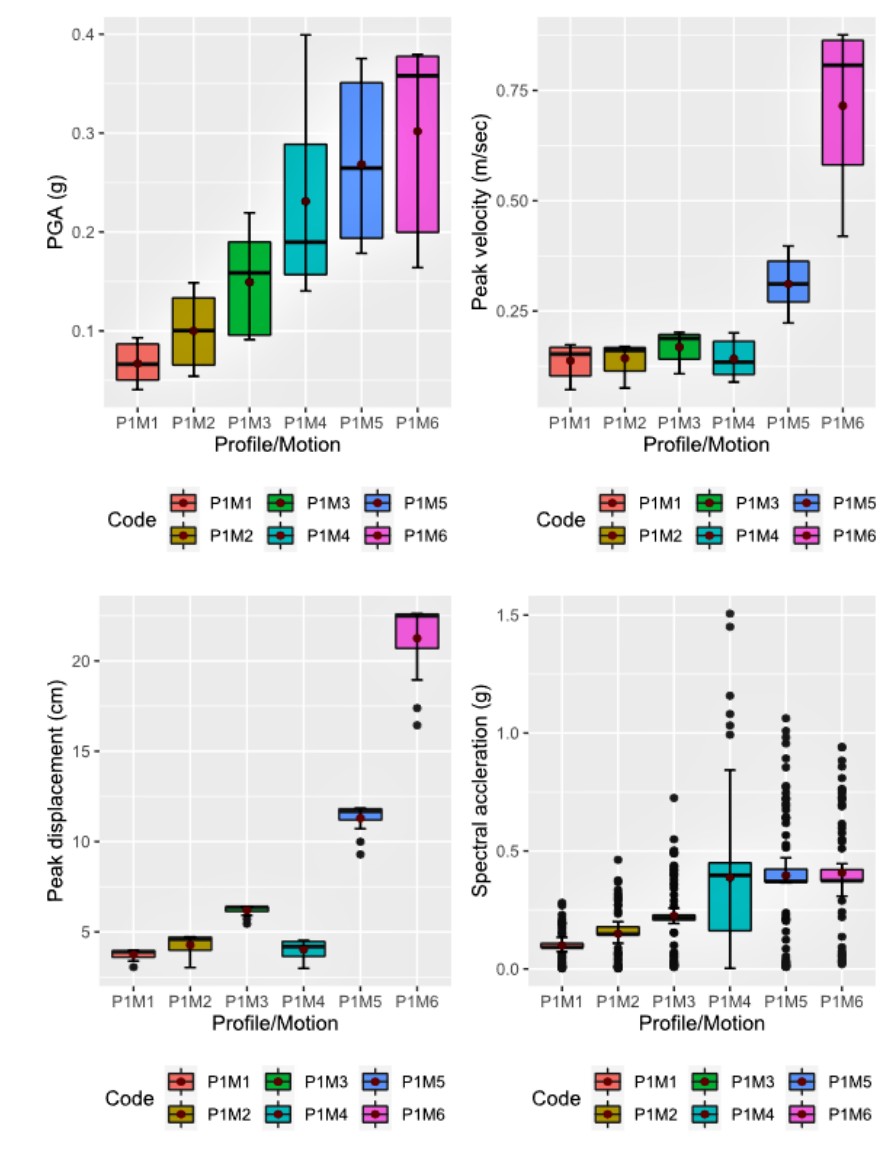

402

**Figure 20:** Box and Whisker plot for ground motion parameters of soil profile at P1 Toorsa II in Zone I.

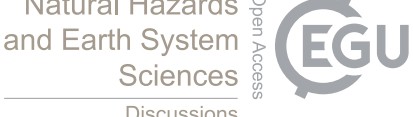

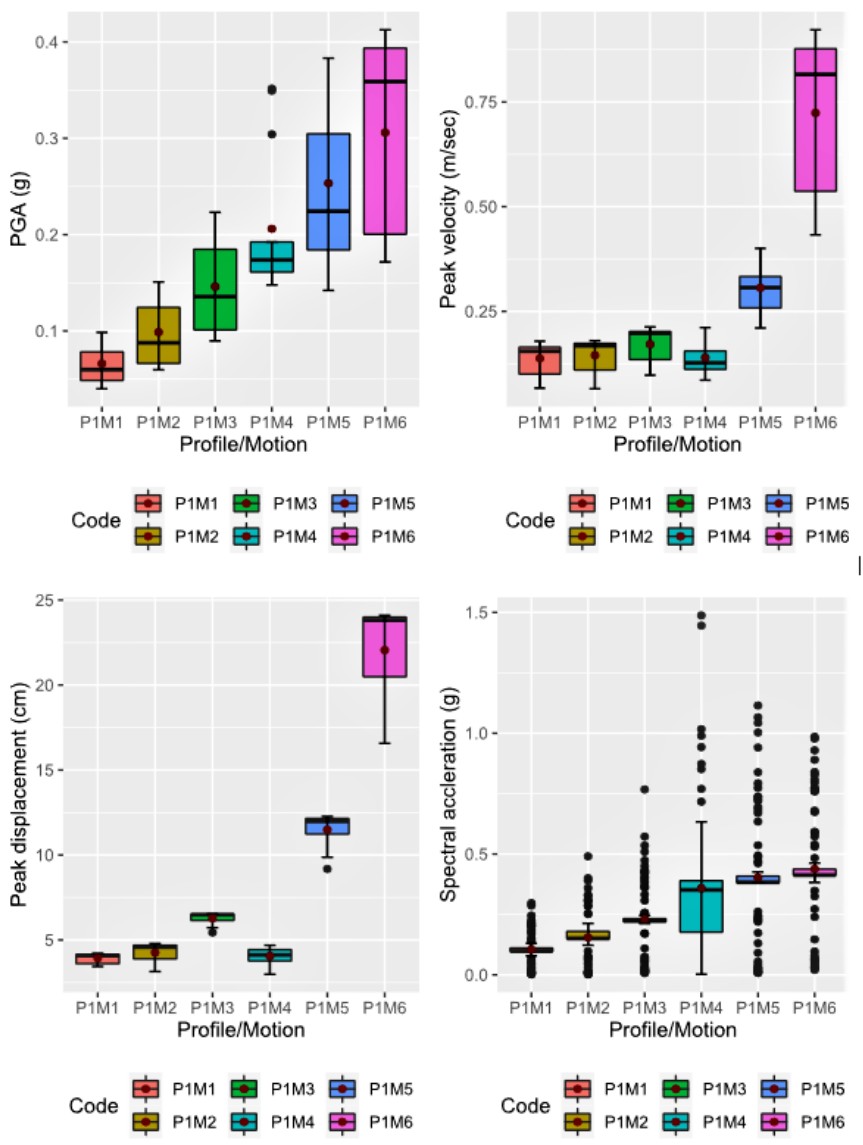

404

**Figure 21:** Box and Whisker plot for ground motion parameters of the soil profile at P1 CST Football Ground in Zone II.

The amplification factor as a function of the correlation matrix is plotted in Fig. 22 for the corresponding input motion M1 to M6 in ascending order of the earthquake intensity (PGA). The plot indicates the existence of a divergent correlation. The moderate ranges of earthquakes are more strongly correlated in terms of the sensitivity of the input motion, e.g., M3 to M5.


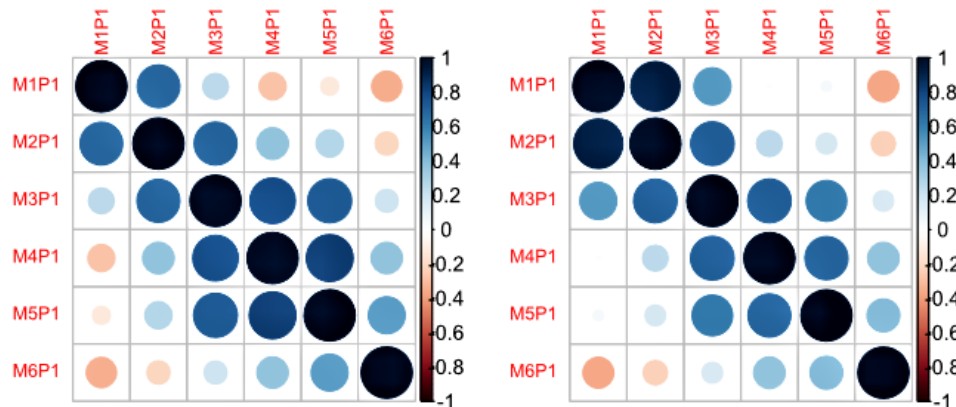

412

**Figure 22:** Correlation matrix for amplification factors as a function of spectral response of soil condition for Zone I (Toorsa II) and Zone II (CST Football Ground) for various input ground motions.

## 5    Conclusions

This study shows the sensitivity of the various input motions using a  1D seismic response analysis. The overall significance of this study can be concluded as follows.

- The trend in the variation of ground motion parameters such as PGA, PGD, PGV, and SA, as expected, projects an increasing order in terms of the intensity of the ground vibration. However, the uncertainty for each parameter is widely scattered, indicating the importance of the variability due to local soil conditions.
- The surface PGA in the investigation area of site classification type B shows about 0.1 g to 0.15 g for the earthquake of low intensity, 0.23 g to ~ 0.38 g for the earthquake of medium intensity, and more than 0.43 g for an earthquake of higher intensity earthquakes such as the 1992 Petrolia earthquake. The result show a higher spectral acceleration of the soil profile in a period range from 0.3 s to 3.0 s with approximately 0.14 g to 1.62 g peak spectral acceleration.
- The critical range of the fundamental natural period is roughly between 0.9 s to ~ 5.0 s with the highest range of seismic wave amplification being between ~ 2.8 to 6.2. In Phajoding, the significance of amplification is comparatively less at ~ 1.7 between 0.4 s and 1.0 s due to a much stiffer soil deposit ($V_{s,30}$ = 584.76 m/s) and a shallow engineering bedrock at 150 m. This suggests that the low-rise buildings are more vulnerable and can see stronger vibrations than the high-rise buildings in Phajoding, however, overall effects on tall buildings cannot be neglected.
- In the present seismic response analysis, the ground response of various strong ground motions with varying ground shaking intensity as an input motion show some anomalies to local soils and proper characterization is recommended when the input motions are selected, especially while performing a site-specific seismic analysis.


**Appendix A: Surface motion parameters**
**Annotations**
P = Profile number
M = Motion number
$G_{rms}$ = Root mean square acceleration
$t_a$ = Bracketed duration
D = Significant duration
CAV = Cummulative absolute velocity
SA = Spectral acceleration
StDev = Standard deviation
**Table A1.** Additional surface motion parameters in Zone I.

| P | M | PV (m/s) | PD (m) | $G_{rms}$ (g) | $t_a$ (s) | D5-95 (s) | D5-75 (sec) | CAV (g-s) | SA @ 1.0 s (g) |
|---|---|---|---|---|---|---|---|---|---|
| 1 | 1 | 0.173 | 0.040 | 0.022 | 1.360 | 7.980 | 2.660 | 0.209 | 0.096 |
| 1 | 2 | 0.170 | 0.048 | 0.033 | 6.000 | 9.080 | 4.100 | 0.318 | 0.181 |
| 1 | 3 | 0.201 | 0.064 | 0.043 | 15.580 | 28.740 | 10.200 | 1.116 | 0.214 |
| 1 | 4 | 0.200 | 0.046 | 0.100 | 14.160 | 8.530 | 4.450 | 0.934 | 0.166 |
| 1 | 5 | 0.397 | 0.120 | 0.071 | 29.340 | 24.580 | 10.420 | 1.580 | 0.649 |
| 1 | 6 | 0.876 | 0.230 | 0.090 | 16.460 | 12.100 | 5.890 | 1.055 | 0.763 |
| 2 | 1 | 0.175 | 0.040 | 0.024 | 1.380 | 7.760 | 2.620 | 0.216 | 0.099 |
| 2 | 2 | 0.169 | 0.047 | 0.035 | 6.000 | 9.060 | 4.020 | 0.328 | 0.192 |
| 2 | 3 | 0.213 | 0.064 | 0.043 | 29.280 | 28.760 | 10.520 | 1.134 | 0.231 |
| 2 | 4 | 0.210 | 0.046 | 0.099 | 14.150 | 8.520 | 4.440 | 0.914 | 0.182 |
| 2 | 5 | 0.419 | 0.119 | 0.074 | 29.340 | 24.580 | 10.360 | 1.625 | 0.727 |
| 2 | 6 | 0.877 | 0.228 | 0.097 | 16.440 | 11.920 | 5.910 | 1.136 | 0.908 |
| 3 | 1 | 0.170 | 0.039 | 0.023 | 1.380 | 7.980 | 2.720 | 0.210 | 0.093 |
| 3 | 2 | 0.164 | 0.047 | 0.034 | 6.000 | 9.080 | 4.100 | 0.323 | 0.176 |
| 3 | 3 | 0.197 | 0.064 | 0.044 | 29.280 | 28.040 | 9.820 | 1.139 | 0.210 |
| 3 | 4 | 0.204 | 0.046 | 0.106 | 14.160 | 8.490 | 4.430 | 0.977 | 0.162 |
| 3 | 5 | 0.414 | 0.118 | 0.073 | 29.340 | 24.580 | 10.440 | 1.626 | 0.645 |
| 3 | 6 | 0.855 | 0.223 | 0.090 | 16.460 | 12.060 | 5.850 | 1.058 | 0.767 |
| 4 | 1 | 0.203 | 0.095 | 0.024 | 0.980 | 7.820 | 2.580 | 0.219 | 0.108 |
| 4 | 2 | 0.201 | 0.065 | 0.034 | 6.000 | 9.380 | 3.880 | 0.326 | 0.204 |
| 4 | 3 | 0.238 | 0.072 | 0.042 | 29.340 | 29.300 | 10.540 | 1.125 | 0.243 |
| 4 | 4 | 0.219 | 0.051 | 0.097 | 12.200 | 8.170 | 4.080 | 0.870 | 0.180 |


| 4 | 5 | 0.417 | 0.135 | 0.065 | 25.900 | 24.580 | 10.180 | 1.455 | 0.685 |
| 4 | 6 | 0.941 | 0.282 | 0.087 | 14.800 | 12.150 | 6.000 | 1.028 | 0.712 |
| Mean | | 0.346 | 0.097 | 0.060 | 15.222 | 15.135 | 6.259 | 0.872 | 0.358 |
| Median | | 0.283 | 0.078 | 0.053 | 10.276 | 13.150 | 5.545 | 0.703 | 0.269 |
| StDev | | 0.256 | 0.071 | 0.029 | 10.088 | 8.330 | 3.021 | 0.472 | 0.271 |
| 84th Percentile | | 0.509 | 0.146 | 0.090 | 30.026 | 22.042 | 9.121 | 1.450 | 0.570 |
| 16th Percentile | | 0.157 | 0.042 | 0.031 | 3.517 | 7.845 | 3.371 | 0.341 | 0.127 |


**Table A2.** Additional surface motion parameters in Zone II.

| P | M | PV (m/s) | PD (m) | $G_{rms}$ (g) | $t_a$ (s) | D5-95 (s) | D5-75 (s) | CAV (g-s) | SA @ 1.0 sec (g) |
|---|---|---|---|---|---|---|---|---|---|
| 1 | 1 | 0.181 | 0.043 | 0.024 | 1.380 | 7.840 | 2.600 | 0.215 | 0.104 |
| 1 | 2 | 0.182 | 0.048 | 0.035 | 6.000 | 8.960 | 4.000 | 0.325 | 0.197 |
| 1 | 3 | 0.215 | 0.066 | 0.043 | 29.320 | 28.800 | 10.480 | 1.135 | 0.231 |
| 1 | 4 | 0.214 | 0.047 | 0.097 | 12.200 | 8.270 | 4.170 | 0.879 | 0.179 |
| 1 | 5 | 0.404 | 0.124 | 0.072 | 29.360 | 24.600 | 10.380 | 1.598 | 0.677 |
| 1 | 6 | 0.936 | 0.244 | 0.095 | 16.470 | 12.060 | 5.770 | 1.106 | 0.816 |
| 2 | 1 | 0.186 | 0.041 | 0.024 | 1.380 | 7.760 | 2.540 | 0.212 | 0.103 |
| 2 | 2 | 0.186 | 0.046 | 0.035 | 5.980 | 8.940 | 3.900 | 0.322 | 0.199 |
| 2 | 3 | 0.217 | 0.067 | 0.042 | 19.000 | 28.900 | 10.520 | 1.101 | 0.232 |
| 2 | 4 | 0.211 | 0.047 | 0.090 | 12.190 | 8.300 | 4.190 | 0.815 | 0.177 |
| 2 | 5 | 0.393 | 0.126 | 0.070 | 29.340 | 24.580 | 10.340 | 1.557 | 0.686 |
| 2 | 6 | 0.943 | 0.250 | 0.096 | 16.450 | 11.920 | 5.800 | 1.101 | 0.839 |
| 3 | 1 | 0.158 | 0.037 | 0.021 | 1.360 | 8.060 | 3.020 | 0.202 | 0.078 |
| 3 | 2 | 0.149 | 0.045 | 0.031 | 6.000 | 9.360 | 4.520 | 0.317 | 0.152 |
| 3 | 3 | 0.178 | 0.062 | 0.044 | 17.040 | 27.420 | 9.720 | 1.116 | 0.175 |
| 3 | 4 | 0.182 | 0.043 | 0.111 | 16.880 | 8.640 | 4.610 | 1.056 | 0.135 |
| 3 | 5 | 0.406 | 0.112 | 0.076 | 29.340 | 24.400 | 10.720 | 1.690 | 0.551 |
| 3 | 6 | 0.830 | 0.218 | 0.092 | 18.050 | 11.900 | 5.390 | 1.103 | 0.704 |
| 4 | 1 | 0.184 | 0.041 | 0.023 | 0.960 | 7.800 | 2.580 | 0.209 | 0.101 |
| 4 | 2 | 0.183 | 0.048 | 0.034 | 6.000 | 8.940 | 3.960 | 0.319 | 0.195 |
| 4 | 3 | 0.212 | 0.066 | 0.041 | 18.960 | 28.840 | 10.520 | 1.084 | 0.227 |
| 4 | 4 | 0.209 | 0.047 | 0.091 | 12.200 | 8.300 | 4.190 | 0.832 | 0.175 |
| 4 | 5 | 0.391 | 0.125 | 0.069 | 29.340 | 24.580 | 10.340 | 1.530 | 0.672 |
| 4 | 6 | 0.905 | 0.243 | 0.091 | 16.440 | 11.990 | 5.870 | 1.056 | 0.793 |
| Mean | | 0.344 | 0.093 | 0.060 | 14.652 | 15.048 | 6.255 | 0.870 | 0.350 |
| Median | | 0.278 | 0.074 | 0.053 | 10.022 | 13.070 | 5.537 | 0.698 | 0.261 |
| StDev | | 0.263 | 0.071 | 0.029 | 9.536 | 8.295 | 3.052 | 0.480 | 0.268 |


| | | | | | | | |
|---|---|---|---|---|---|---|---|
| 84th Percentile | 0.506 | 0.140 | 0.090 | 28.804 | 21.920 | 9.109 | 1.451 | 0.559 |
| 16th Percentile | 0.153 | 0.039 | 0.031 | 3.487 | 7.793 | 3.365 | 0.335 | 0.122 |


**Data availability**

All the data used in this study are presented in the paper.
**Author contribution**
Conceptualization (KT), Data curation (KT), Formal analysis (KT), Funding aqisition (KRA), Methodology (KT,
DG and GF), Resources (KT, DG and KRA), Software and visulization (KT), Writing – original draft preparation
(KT), Writing – review & editing (DG, NC, GF and KRA).
**Competing interests**
The authors declare that they have no competing interests.
**Acknowledgements**
The authors are thankful to Phuentsholing Thromde (Municipal office) for providing additional geotechnical data.

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
