# Peer review of "Sensitivity analysis of input ground motion on surface motion"

_Natural Hazards and Earth System Sciences, 2021_

## Author Response (AR1)

**Reviewer #1**

General comments

This paper presents the first attempt of local seismic effects assessment in Bhutan. Considering the small amount of input data available (i.e., no instrumental records of past earthquakes, reduced geotechnical characterization of soil deposits, etc..) this study represents a first and relevant step towards a possible future deeper seismic hazard assessment. However, several critical problems need to be addressed and explained to let the manuscript be accepted for publication.

Specific comments

1. Introduction: This manuscript focuses on the study of seismic hazard more in particular on seismic site effects, however few lines and only one reference citation on site effects is reported in the introduction. A wider review of past studies focused on this topic needs to be integrated in the manuscript.

   *Author Response: We appreciate the reviewer's constructive feedback. We selected several other literatures from relatively similar geo-tectonic settings.*

2. Lines 113 – 117: This sentence does not seem to be consistent with Figure 2 since the geological formations falling in the study area (black square in Figure 2) are not the same reported in the sentence.

   *Author Response: Changed accordingly.*

3. Authors performed a reconstruction of the groundwater table in the study area considering 29 borehole data without indicating the season during which the data has been recorded. Reasonably, the groundwater table position varies during the year. An evaluation of the possibility of grouping borehole by seasons could help to refine the map, particularly in the area between Dhamdhara, Pipaldara abd Kabreytar and in that close to Rinchending.

   *Author Response: Thanks for pointing this out. We have now added description about the groundwater table.*

4. Lines 258 -259: "[..] low, medium, and high […]" with respect to what? Please specify in text

   *Author Response: We used low, medium, and high to indicate the PGA range generically. Meaning,*

5. Authors reported in Figure 11 the Fourier Spectra of the considering earthquake as Fourier Amplitude vs Period. It is common use to represent Fourier Spectra as Fourier amplitude vs frequency, so this representation confuses the reader. It is opinion of this reviewer that just the X label is incorrect, but please check this figure and modify it consequently.

   **Response: We updated the figure accordingly.**

6. Figure 12: This figure presents the variation of PGA induced by each earthquake at different depths in the soil. It could be more useful to present data by normalizing them to the maximum PGA of the earthquake input. Moreover, in case of Zone II the bedrock depth is fixed at 400m so Figure 12b should present data up to this depth.

Author response: Thanks for the insightful comment. We updated the figures accordingly.

7. Authors performed a series of 1D linear-equivalent numerical modelling of eight soil columns representative of the study area and reported the results in Figures 13 and 14. They showed the response spectra at bedrock and on the surface. While results obtained by applying earthquake from M1 to M4 seem to be consistent, those obtained by considering M5 and M6 look anomalous. Furthermore, in the latter case the response spectra at the bedrock level are characterized by anomalous peaks at low period that are completely nullified at the ground level. I suggest the authors to check the signal processing of these earthquakes (M5 and M6) and verify the consistency with the input applied in the numerical simulations.

**Author response: Thank you for pointing out this. We revisited analysis and explained the parameters in the revised manuscript.**

8. All the presented results need to be more deeply discussed. Moreover, considering the shaking level of the seismic input and the typology of numerical simulation, the topic of non-linear behavior of the soil material should be addressed. This could also help for a better interpretation of the results (i.e. Figures 17 and 18)

**Author response: We appreciate the reviewer's comment. We actually restructured and improved the manuscript. We have provided related discussions in the revised version of the manuscript.**

Technical correction

1. Line 24 -26: This sentence is not clear

**Author response: Rectified.**

2. Line 86: Please specify which site effect you are investigating

**Author response: The current study in particular draws site effect in terms of amplification factor attributed mainly due to input motion amplitude parameters.**

3. Line 186: This sentence is not correct.

**Author response: Rectified.**

4. Line 189: This in-text citation is not present in the reference list

**Author response: We have updated citation and reference list.**

Lines 359 - 360: This sentence is not clear.

Author response: Rectified.

5. Line 396 – 398: This sentence is not clear.

**Author response: Rectified.**

6. Are Baxa (Figure 2) and Buxa (Line 113) the same lithological group?

**Author response: Buxa is correct lithological group. Corrected in the revised manuscript.**

7. Table 1: What "-do-" stands for?

**Author response: The authors intend to indicate same test methods in Dhamdhara and Rinchending. It is removed in the revised manuscript.**

8. Figures 1b: The legend is missed

**Author response: Legends modified accordingly.**

9. Figure 2: Please add the location of boreholes reported in Figure 5. Moreover, north direction and scale are missed.

**Author response: Borehole locations are added in the figure.**

10. Line 173 – 175: Acronyms should be explicitly reported in the manuscript.

**Author response: Acronyms list is provided in the manuscript.**

11. Line 195-197. This sentence about liquefaction and corresponding potential is out of the paper topic. Please delete it.

**Author response: Removed.**

12. Figure 3: The legend is not clearly legible

**Author response: Modified accordingly.**

13. Figure 7: The resolution of this figure is too low

**Author response: Resolution of the figure is improved.**

14. Figure 10 – 11: To improve the manuscript readability, these figures could be merged in a unique figure composed of two columns, one devoted to time histories and another to the corresponding FFT spectra.

**Author response: Related figures are merged.**

15. Figure 12: Colors chosen for Earthquakes M4, M5 and M6 are M6 are too similar. Please use more distinguishable colors

**Author response: Modified as suggested.**

16. Figure 13: Please add "bedrock" and "ground surface" as labels in the graph

**Author response: Added as suggested.**

17. Figures 15 -16: Please specify how you calculated the red line FFT. Is it an average of the FFT values at each time step? Please specify in the text.

**Author response: The red line illustrated in the graph is an average value.**

18. Figure 17b/d – 18b/d: How have you calculated the "Response spectrum intensity" and "Mean frequency"? Please specify in the text

**Author response: Response spectrum intensity is used to indicate Arias intensity, mean frequency is frequency.**

19. All the figure's caption should be improved

**Author response: Almost all figures are replotted to improve quality of plot and caption.**

20. In-text citations need to be modified according to the journal guidelines.

**Author response: Modified.**

Reviewer #2

Authors have presented a research on Sensitivity analysis of input ground motion on surface motion parameters in high seismic regions: A case of Bhutan Himalaya. This paper is most like a lecturing note has serious drawbacks in both poor presentations and applied technics. This paper deals with a very known approach without any new contribution. A critical drawback of this work is that the reliability of the collected geotechnical data was not comprehensively reviewed. The paper is well structured and written, however, considering the quality of the paper and uniqueness of the research, I have concluded that the manuscript is not suitable for publication in Natural Hazards and Earth Science Systems.

- Abstract and introduction are very general and there is no significant in the scientific border. There is ample room to improve the introduction as "Introduction" is actually weak.

**Author response: We have updated the entire manuscript.**

- Presenting facts is not sufficient for a journal paper, there needs to be more direction to the writing and evidence of critical analysis.

**Author response: We have critically discussed the findings.**

- Authors highlighted anomalous damage patterns in various parts of Bhutan due to the earthquake but failed to present any data or photos etc. to highlight the damage.

**Author response: We do not have such pictures but some records of news outlets. We used the same to write this sentence.**

- Some of the statements presented in the manuscripts are contrary to each other.

e.g. Abstract:....this study is the first attempt to quantify the influence of the 14 local site conditions in the eastern fringe of the Himalaya...

Introduction: In Bhutan, very few studies on local seismic response analysis have been 34 conducted so far. Some of the recent studies

**Author response: The abstract and introduction sections are totally restructured. The revised manuscript has consolidated literature review too.**

There are many others..........

- I do not see any earthquake greater than Mw 7.0 in Figure 1, but authors mentioned that records of past earthquakes in Bhutan are available since 1713 (Mw 7.0). Authors highlighted at several places that Bhutan is one of the most seismically active regions in the world.

**Author response: According to literature review, the statement "earthquake greater the Mw 7.0 since 1973" has been indicative. However, no accelerograms are available for this earthquake.**

- Page 2, Line 49: Local site conditions during historical earthquakes in Bhutan were identified as the main cause of structural damage. Any documents or reference or photos ?

**Author response: The authors intended to mention damages being caused due to its geographical and geological settings. The sentence will be revised to align the aim of the current study. The damages of structures refer to the recent earthquake of April 2020 and September 2009 earthquake that are not documented in the same fashion as stated. Thus we inferred this sentence based on our own interpretation.**

- Page 5, Lines 129-140: Is this paragraph related to Seismicity and geological setting of the study area? There are some information which seem to be irrelevant for the article.

**Author response: Corrected.**

- Page 6: The groundwater table in the study area is shallow and varies between 0.5 m to 16.0 m. Are you still considered groundwater table at 16 meter as shallow depth? Please double check literature.

**Author response: The groundwater table variation in the recorded boreholes is fetched from geotechnical report. We confirmed with the original report.**

- Page 10, Line 195-196. The term liquefaction came out of nowhere.

**Author response: The statement about liquefaction is deleted.**

- What about temporal variation of water table? Groundwater table is highly sensitive to rainfall. Did you consider it? It seems authors consider the groundwater table at the time of geotechnical investigation which can be random throughout the year. This could be attributed to wide range of groundwater table within small area.

**Author response: The investigation date is available. However, details are limited. Thus, we have used the figure that is as illustrative as possible.**

- Table 1 shows more than 100 soil samples, but authors presented only 7 particle size distribution curves. Where is PSD curve for Rinchending?

**Author response: The authors presented average particle size distribution of each site for holistic representation of particle size. The Figure 6 (Zone II) is the Rinchending area which consists of four sites.**

- Table 1: The dry density of soil sample is higher than the bulk density for Rinchending. Even I do not believe the cohesion of the sample having SPT-N value higher than 100 is only 0.18 $kg/cm^2$. Where is plastic limit and liquid limit values?

**Author response: We checked and confirmed the provisions.**

- Soil at Dhamdhara is described as coarse-grained sand with gravel/cobbles and rock piece but authors reported PI value. PI values at Rinchending and Dhamdhara is not reliable based on soil descriptions.

**Author response: We used the values provided in the report. However, we cannot guarantee the reliability of the values since we are not the ones who did all the tests.**

- Double digit for shear wave velocity is not required.

**Response: Modified.**

- Quality of Figure 7 can be improved significantly.

  **Response: Redrawn.**

- Authors should present the profile of peak ground acceleration (PGA) for the locations presented in Figure 5.

**Author response: Provided accordingly.**

- The manuscript currently lacks a cogent argument / thread. This stems from the introduction, which lacks an aim.

  **Author response: We have maintained coherence in the modified version.**

- Could the findings in the study be applied to any countries or only adopt to Bhutan? Please describe the contribution of this study from the viewpoint of local characteristics and universal ones.

**Author response: The conceptual framework remains valid globally for different soil conditions.**

- Why not consider one or two ground motion from earthquake in Himalaya?

**Author response: Since there are no records from Bhutan Himalaya, we selected overseas earthquakes.**

- Irrelevant self citation. Similarity index is quite high. Some of the text were copied exactly from the following paper and other reports.

Tempa et al. (2021). "Shear wave velocity profiling and ground response analysis in Phuentsholing, Bhutan" , Innovative Infrastructure Solutions.

**Author response: The entire manuscript is restructured.**

- Page 17, Lines 282-287: This is like a lecture note, a very well-known statement.

**Author response: Rectified.**

- Figure 12, why not PGA profile is presented up to bedrock? I would suggest to present PGA profile in terms of amplification ratio or factor by normalized PGA of input motion. It will help to visualize the results.

**Author response: Figures are updated in the revised version.**

---

## Author Response (AR2)

05/07/22

The editor,

Natural Hazards and Earth System Sciences, Copernicus.

Thank you for providing us with notable remarks. The constructive comments from the special issue editor and reviewers have greatly improved the quality of the manuscript. We addressed the issues raised by the editor as follows:

Editor comments:

The paper has been thoroughly revised by the authors based on the changes proposed by the reviewers. We gave the authors the opportunity to do a major revision, despite the fact that one of the reviewers proposed to reject the paper, and I think that was a good choice, because the authors were able to greatly improve the paper and to address the criticisms raised by the reviewer.
There are still two issues that I would like to see addressed by the authors before publication.
1) although I am not a native English speaker, I have a clear feeling that the English is not as fluent as it should be and that it is not always precise and elegant. I would ask for an effort from the authors to further improve their English, perhaps with the support of a native speaker. The reading of the article would benefit greatly.

*Author response: Thank you for raising an important concern. We have now thoroughly corrected manuscript language. Please see attached track changed version.*

2) I am not convinced by the conclusions. As often happens, more than conclusions they seem to me a summary for points of the discussion. If possible, I ask for an effort to elaborate a more prospective conclusion, maybe highlighting the importance of the work for seismic hazard. The authors need to remember that NHESS is not a specialist seismic journal, but welcomes a broad community of researchers from other fields who are not familiar with technical aspects of applied seismology. Therefore, it would be appropriate for the conclusions to also address this audience by highlighting the importance of the findings to hazard in general, and reiterating what is new in this work.

*Author response: The editor has provided us with very crucial feedback. We also agree on the fact that the conclusions were not really good in the earlier version of the manuscript. In the revised version, we rewrote the conclusions considering suggestions from the editor. We have also interlinked our findings with broad seismic hazard and preparedness perspectives.*

We look forward to receiving your feedback.

Sincerely,

Dipendra Gautam

Corresponding author